# Phase-tuned neuronal firing encodes human contextual representations for navigational goals

Andrew J Watrous[1,2,3]*, Jonathan Miller[1], Salman E Qasim[1], Itzhak Fried[4,5,6,7], Joshua Jacobs[1]*

[1]Department of Biomedical Engineering, Columbia University, New York, United States; [2]Department of Neurology, Dell Medical School, University of Texas at Austin, Austin, Texas, United States; [3]Seton Brain and Spine Institute, Austin, Texas, United States; [4]Department of Neurosurgery, David Geffen School of Medicine, University of California, Los Angeles, Los Angeles, United States; [5]Semel Institute for Neuroscience and Human Behavior, University of California, Los Angeles, Los Angeles, United States; [6]Sackler Faculty of Medicine, Tel-Aviv University, Tel-Aviv, Israel; [7]Department of Neurosurgery, Tel-Aviv Medical Center, Tel-Aviv, Israel

**Abstract** We previously demonstrated that the phase of oscillations modulates neural activity representing categorical information using human intracranial recordings and high-frequency activity from local field potentials (Watrous et al., 2015b). We extend these findings here using human single-neuron recordings during a virtual navigation task. We identify neurons in the medial temporal lobe with firing-rate modulations for specific navigational goals, as well as for navigational planning and goal arrival. Going beyond this work, using a novel oscillation detection algorithm, we identify phase-locked neural firing that encodes information about a person's prospective navigational goal in the absence of firing rate changes. These results provide evidence for navigational planning and contextual accounts of human MTL function at the single-neuron level. More generally, our findings identify phase-coded neuronal firing as a component of the human neural code.

DOI: https://doi.org/10.7554/eLife.32554.001

*For correspondence:
andrew.j.watrous@gmail.com (AJW);
joshua.jacobs@columbia.edu (JJ)

**Competing interests:** The authors declare that no competing interests exist.

## Introduction

Single-neuron firing forms a fundamental basis of the neural code during perception and memory. In addition to the well-established role for behavior-related changes in neuronal firing rates, converging evidence across species and behaviors suggests that interactions between single-neuron spike timing and oscillations observed in the local field potential (LFP) also contribute to the neural code (*Hyman et al., 2005*; *Huxter et al., 2003*; *Rutishauser et al., 2010*; *Belitski et al., 2008*; *Ng et al., 2013*; *Kayser et al., 2009*). In rodents, hippocampal and medial prefrontal cells show phase precession relative to theta oscillations during navigation (*O'Keefe and Recce, 1993*; *Terada et al., 2017*; *Jones and Wilson, 2005*), in which the theta phase of neuronal firing represents information about the animal's position (*Jensen and Lisman, 2000*).

These observations have been incorporated into theoretical models of neural coding that hypothesize a general role for oscillatory phase for coding various types of behavioral information (*Nadasdy, 2009*; *Kayser et al., 2012*; *Lisman and Jensen, 2013*; *Watrous and Ekstrom, 2014*). For example, in Spectro-Contextual Encoding and Retrieval Theory (SCERT), we proposed that frequency-specific and phase-locked neuronal firing to low-frequency oscillations at different phases (i.e. phase coding) also forms a basis of the human neural code (*Watrous and Ekstrom, 2014*;

*Watrous et al., 2015b*). We previously reported evidence for SCERT (*Watrous et al., 2015a*) using high-frequency activity in the LFP as a proxy for single-cell spiking (*Crone et al., 1998*; *Manning et al., 2009*; *Miller et al., 2014*). However, given the complex and variable relationship (*Ekstrom et al., 2007*; *Manning et al., 2009*; *Rey et al., 2014*) between the spiking of particular single neurons and high-frequency activity in the human medial temporal lobe (MTL), it is unclear whether human MTL neurons show phase coding of navigationally relevant information beyond an overall preference to fire at particular phases (*Jacobs et al., 2007*). We thus sought to extend our previous findings of LFP phase coding (*Watrous et al., 2015a*) to the single-neuron level in patients performing a virtual navigation task, hypothesizing that phase coding would occur to low-frequency oscillations based on both human studies (*Jacobs et al., 2010*; *Watrous et al., 2011*; *Ekstrom et al., 2005*; *Mormann et al., 2008*) and the above-described rodent work.

An optimal navigator must both plan routes and recognize when they have arrived at their destination. Human imaging and lesion evidence indicate that activity in the human MTL and medial prefrontal cortex forms active representations of spatial context such as navigational goals (*Ranganath and Ritchey, 2012*; *Brown et al., 2016*; *Ciaramelli, 2008*; *Spiers and Maguire, 2007*; *Wolbers et al., 2007*) in support of navigational planning (*Horner et al., 2016*; *Bellmund et al., 2016*; *Kaplan et al., 2017*). Analyzing human single-neuron recordings from the MTL, previous studies have identified neurons that increase their firing rate when viewing goal locations (*Ekstrom et al., 2003*). To date, it is unclear whether phase-coding also exists for navigational goals. It is also unknown whether rate and phase-coding co-exist in humans, as suggested by rodent studies that indicated that phase coding was a distinct phenomenon compared to rate coding (*Huxter et al., 2003*; *Hyman et al., 2005*).

Drawing upon the phase-coding hypotheses from SCERT and related findings in rodents (*Hollup et al., 2001*; *Hok et al., 2007*; *Hyman et al., 2005*; *O'Neill et al., 2013*), we hypothesized that spatial contextual representations for specific navigational goals would be implemented by distinctive patterns of phase coding by individual neurons. Moreover, based on rodent (*Wikenheiser and Redish, 2015*) and human studies (*Viard et al., 2011*; *Howard et al., 2014*; *Brown et al., 2016*; *Horner et al., 2016*; *Bellmund et al., 2016*) implicating medial temporal lobe structures and frontal cortex in navigational planning, we reasoned that spike-phase coding may support these behaviors at the single-neuron level, hypothesizing that distinctive spike phase patterns would correspond to the neural network states representing planning and searching for particular goals. SCERT generally predicts that oscillatory frequencies should match between encoding and retrieval and that phase coding should occur at the dominant oscillatory frequency that occurs in a particular behavior and brain region. Thus, based on the body of evidence indicating hippocampal slow-theta oscillations are the most prominent during human virtual navigation (*Ekstrom et al., 2005*; *Watrous et al., 2011*; *Jacobs, 2014*; *Bush et al., 2017*), we predicted here that phase coding should occur primarily at slow theta frequencies.

To test these ideas, we analyzed a dataset that simultaneously measured human single-neuron and oscillatory activity from MTL (hippocampus, entorhinal cortex, amygdala, and parahippocampal gyrus) and frontal (medial prefrontal/cingulate, motor, orbitofrontal) regions during a goal-directed navigation task (*Figure 1—figure supplement 1*; *Jacobs et al., 2010*; *Miller et al., 2015*). After first assessing changes in firing rate related to goal activity, we then asked if additional goal-related information is encoded by considering oscillatory phase during spiking. Following the analytic strategy from our previous work (*Watrous et al., 2015a*), we tested for frequency-specific phase locking and then directly tested for phase coding, which would appear as individual neurons that spiked at different phases according to the prospective goal. In addition to cells that encode navigational variables using firing rate, our results confirmed the existence of phase coding for navigational goals in individual neurons, thus providing the first evidence for the oscillatory phase coding of spatial contextual information in the human brain.

## Results

### Behavior and neuronal firing during goal-directed navigational planning and arrival

Patients performed a goal-directed navigation task in which they moved throughout a circular environment delivering passengers to one of six goal locations located on the outer edge of the environment (see *Jacobs et al., 2010* for details). Upon arriving at a goal store, the patient paused and then was instructed to navigate to a new goal store. On each trial, patients thus had to make a navigation plan about which direction of movement in the environment would lead them most directly to the location of their goal. Driving time between stores significantly decreased throughout the task session (Kruskal-Wallis test across sessions, p=0.007), indicating that the patients successfully learned the environment and planned efficient paths between stores.

Previous work in humans has identified single neurons responsive to navigational goals (*Ekstrom et al., 2003*) and imaging work suggests that the MTL is involved in navigational planning ( *Bellmund et al., 2016*; *Horner et al., 2016*; *Brown et al., 2016*). We investigated the single-neuron correlates of these phenomena in our task. We assessed neuronal firing rate as a function of the identity of the navigational goal and of different task periods using a two-way ANOVA, with factors for goal and task period ('planning' vs. 'arriving', see Materials and methods). *Figure 1A* shows an example entorhinal neuron who's firing significantly increased during deliveries to goal store 3 (main effect of goal, p<0.0001). We identified 53 such goal-responsive cells (11% of 466 MTL neurons; main effect of goal, p<0.05), which were present in 11 of 12 patients. We observed significant counts of goal-responsive neurons in the hippocampus, entorhinal cortex, orbitofrontal cortex, and premotor cortex (*Figure 1B*; Binomial tests, p's < 0.05).

We also identified cells that showed significantly enhanced firing during either the navigational planning or the arrival period of each trial (main effect of task period, p<0.05). *Figure 1C* shows an example hippocampal neuron whose firing rate significantly increased during navigational planning compared to goal arrival (main effect of task period, two-way ANOVA, p<0.0001, followed by posthoc analysis). We observed significant counts of navigational planning neurons in the hippocampus of 9 of 12 patients and in all areas except the amygdala (*Figure 1E*; Binomial test, p<0.05). Furthermore, we observed modulation of firing rate by arrival at goals in parahippocampal and motor areas (*Figure 1D–E*). We found 24 cells with significant interactions between goal and task period (p<0.05). These results provide single-neuron evidence that the MTL encodes information about navigational goals, and supports navigational planning towards reaching these goals, using modulations in firing rate, extending previous findings (*Ekstrom et al., 2003*; *Watrous et al., 2011*; *Brown et al., 2016*).

### Slow theta oscillations (3 Hz) in the MTL during virtual navigation

Our primary hypothesis was that human MTL neurons encode behavioral information by modulating their spiking based on the phase of slow oscillations beyond changes in firing rate. Examining this hypothesis required that we accurately identify the presence and phase of slow oscillations, particularly because human MTL oscillations are lower frequency and less stationary compared to the stable theta oscillations observed in rodents (*Watrous et al., 2013*; *Vass et al., 2016*). We developed and benchmarked a novel method, the Multiple Oscillations Detection Algorithm ('MODAL'; *Figure 2A–C*), to detect and characterize neural oscillations in adaptively identified band(s) whose frequency ranges are customized for each recording site according to its spectral properties. MODAL identifies narrow-band oscillations exceeding the background 1/f spectrum (*Figure 2A*) and calculates the instantaneous phase and frequency of oscillations in each band (see Materials and methods) while excluding time points without oscillations or that exhibited epileptogenic activity (*Gelinas et al., 2016*). Thus, MODAL allowed us to test for phase coding of spikes during the presence of narrow-band oscillations in our dataset.

MODAL reliably identified oscillations at multiple frequencies that were visible in the raw trace (*Figure 2B–C*). Analyzing each of 385 LFP channels from the MTL across the entire task period using MODAL, we found that most channels showed a band of activity centered at 'slow theta' (~3 Hz; 93% of electrodes; *Figure 2D*, gray line). Analyzing the overall amount of time each frequency was detected on these electrodes, we found that slow theta was detected most often (*Figure 2D*, black

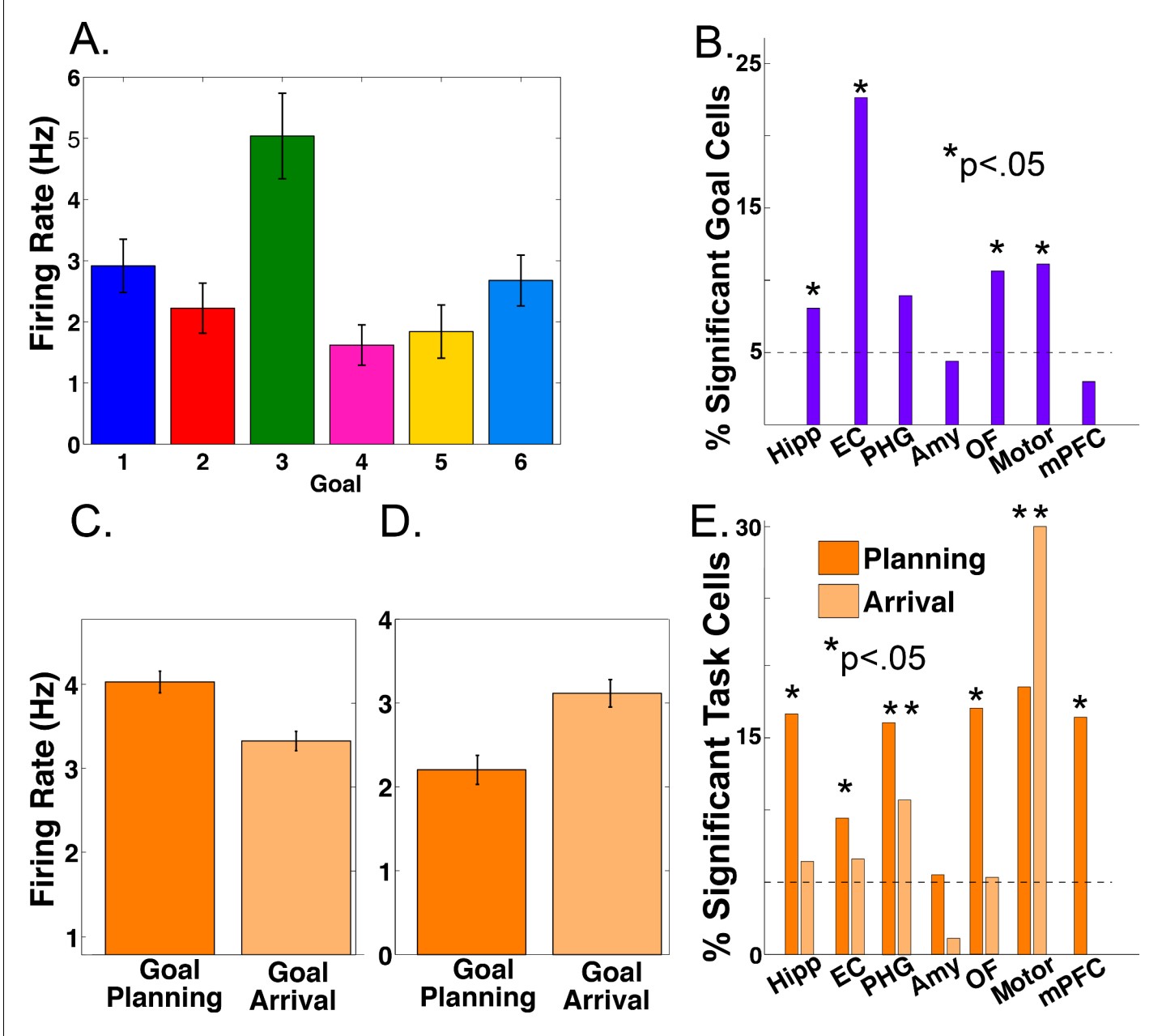

**Figure 1.** Firing rate modulations by navigational goal and task phase. (**A**) Neuron from the entorhinal cortex of patient four whose firing rate was significantly goal-modulated when delivering to goal 3 (p<0.001). Firing rate is plotted as a function of each navigational goal (error bars indicate s.e. m.). (**B**) Proportion of goal-responsive neurons in each brain area. Asterisk indicates significant counts using binomial test. (**C**) Example neuron from the hippocampus of patient 12 whose firing rate was modulated during goal planning (p<0.0001). (**D**) Example neuron from the parahippocampal gyrus of patient eight whose firing rate was modulated during goal arrival (p=0.0002). (**E**) Proportion of task-responsive neurons in each brain area, shown separately for planning and arrival. See methods for region acronyms.

DOI: https://doi.org/10.7554/eLife.32554.002

The following figure supplement is available for figure 1:

**Figure supplement 1.** Task and recording methods.

DOI: https://doi.org/10.7554/eLife.32554.003

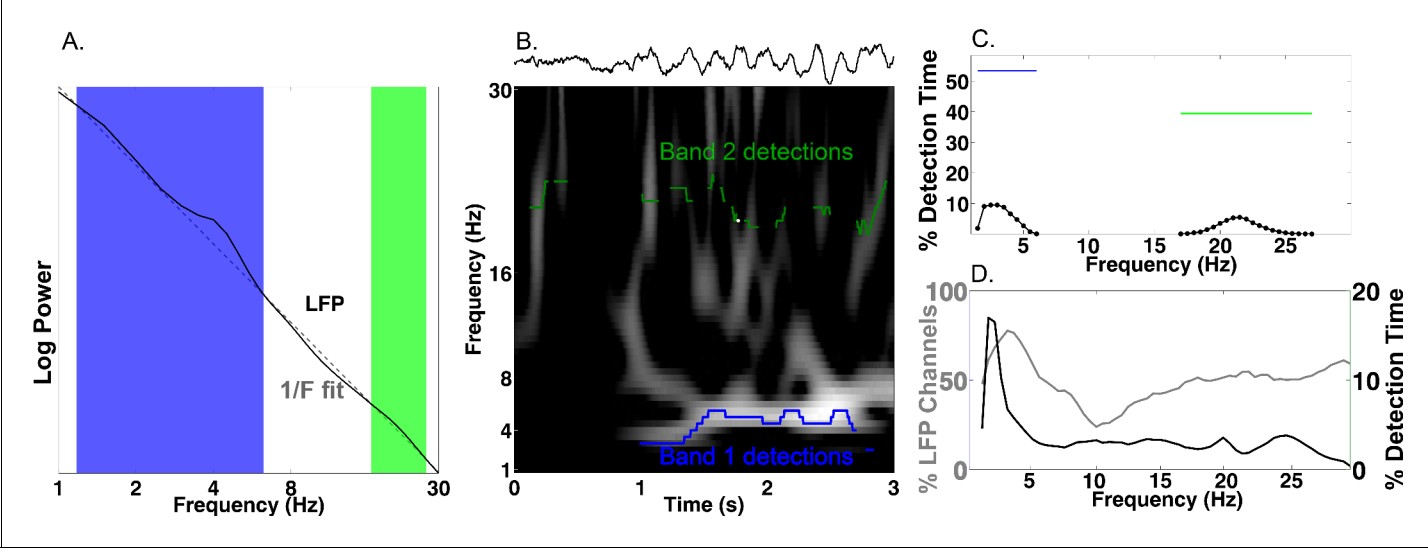

**Figure 2.** Multiple Oscillation Detection Algorithm ('MODAL'). (A-C) Key steps in the algorithm, shown for an example electrode from the right hippocampus of patient 9. (A) Mean log power averaged over time (black) and a fit line of the 1/f background spectrum (gray). A slow theta band (blue) and a beta band (green) are identified as contiguous frequencies exceeding the fit line. (B) Example output from MODAL depicting a raw trace example of the LFP (upper) with the detected oscillations in each band (lower). The instantaneous frequency of the detected oscillation in each band is overlaid on a spectrogram and gray portions of the spectrogram indicate power values exceeding a local fit (similar to A but using a 10 s epoch). (C) Accumulating detections over time reveals the prevalence of oscillations at each frequency on this electrode (black). Blue and green bars indicate the overall prevalence of oscillations in each frequency, independent of the exact frequency within a band. (D) Population data for MTL channels demonstrating low frequency oscillations. Grey line indicates the percent of LFP channels with a detected band as a function of frequency. Of those channels with a detected band, the black line indicates the average amount of time each frequency was detected. Slow theta oscillations (below 5 Hz) are observed using both metrics.

DOI: https://doi.org/10.7554/eLife.32554.004

The following figure supplements are available for figure 2:

**Figure supplement 1.** Proportion of channels with oscillations detected using MODAL in each brain region.

DOI: https://doi.org/10.7554/eLife.32554.005

**Figure supplement 2.** Analysis of rodent CA1 and medial prefrontal cortex LFPs using MODAL.

DOI: https://doi.org/10.7554/eLife.32554.006

line). Similar results were identified in different brain areas (*Figure 2—figure supplement 1*). We then verified that MODAL can capture multiple narrowband oscillatory signals using a published rodent recording dataset (*Fujisawa et al., 2008*; crcns.org PFC-2 dataset), and observed canonical rodent hippocampal CA1 theta oscillations and a more variable low-frequency rhythm in the medial prefrontal cortex (*Figure 2—figure supplement 2*). These results indicate that MODAL is able to identify and track the dynamics of narrowband signals, providing cross-validation for our human findings which are consistent with previous work showing the prevalence of slow theta in the human MTL (*Watrous et al., 2011*, *2013*; *Vass et al., 2016*; *Jacobs, 2014*; *Bohbot et al., 2017*). We subsequently restricted our analysis of phase coding to this low-frequency band (1–10 Hz) because it was most prominently detected by MODAL and because activity in this band has been shown to modulate human single-neuron firing (*Jacobs et al., 2007*).

## Slow theta phase modulates neuronal firing

As a precursor to testing for phase coding, we asked if phase coordinated the activity of individual neurons across the entire task session in the bands identified by MODAL. Focusing first on the MTL, we analyzed 466 neurons that each had a simultaneously recorded LFP with an oscillation in a low-frequency band (1–10 Hz). In many cells we observed significant phase-locking, an overall tendency for firing to increase at particular phases of the LFP oscillation (*Jacobs et al., 2007*; *Rey et al., 2014*). Phase locking is evident by examining the LFP phase distribution for all spikes that occurred during oscillations from a given cell (*Figure 3A* upper panel, Rayleigh p<0.005). Across our

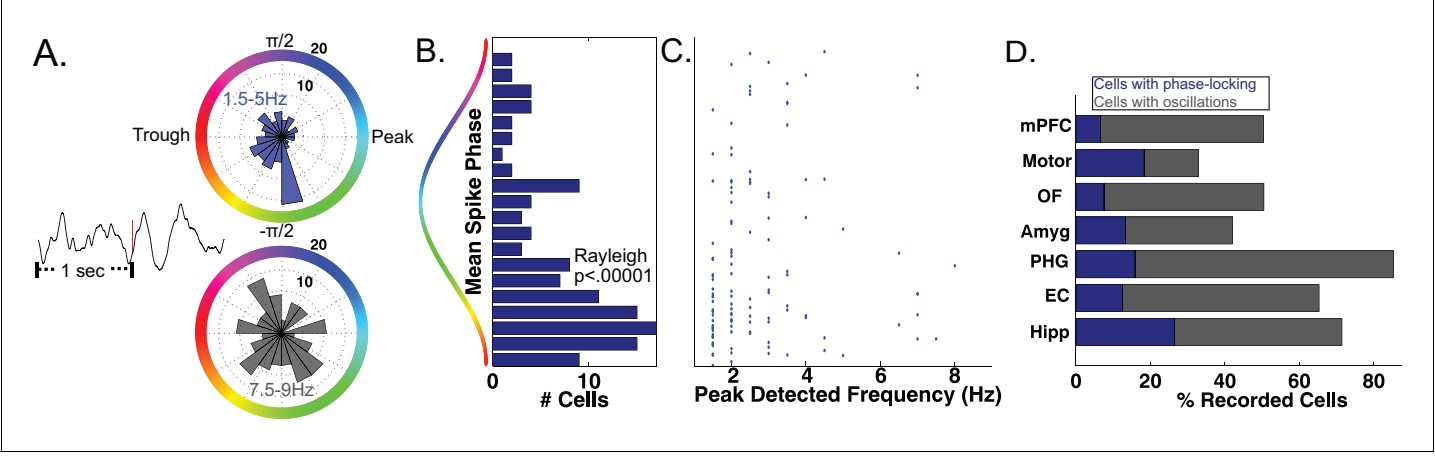

**Figure 3.** Phase-Locked Neural Firing to low-frequency oscillations. (**A**) Spike-triggered average of a phase-locked neuron from the right hippocampus of Patient 1 (left). Red tick mark denotes a spike. Circular histograms (right) show phases at which spikes occurred relative to two detected bands. Spiking was phase-locked to the ascending phase in the 1.5–5 Hz band (red) but not in the 7.5–9 Hz band (Rayleigh test, p=0.004 and p=0.34, respectively). (**B**) MTL Population data: Pooling over frequencies, mean spike phases were significantly clustered near the initial ascending phase of the oscillation (Rayleigh test, p<0.00001). (**C**) Population scatter plot of the mean phase of firing and maximally detected frequency within the band for each phase-locked MTL neurons. (**D**) Population results showing proportion of phase-locked neurons in each brain region. Total bar height indicates the proportion of neurons recorded on an LFP channel with a band in the 1–10 Hz range. See methods for region acronyms.
DOI: https://doi.org/10.7554/eLife.32554.007

population of MTL neurons, we identified phase-locked neural firing in 144 neurons (144/466, 30%, Rayleigh test, p<0.005), a proportion significantly above chance (Binomial p<0.00001). We observed that phase locked neural firing was clustered just after the trough of the oscillation for these cells (*Figure 3B*, Rayleigh test p<0.00001) and most phase locking occurred to slow-theta oscillations below 5 Hz (*Figure 3C*). Significant counts of phase-locked neurons were observed in each brain region (Binomial test, p<0.0001) and we observed phase-locking most prominently in the hippocampus (*Figure 3D*). These results confirm the presence of phase-modulated neuronal activity in this dataset.

The SCERT model predicts that neuronal activity is modulated by oscillations at particular frequencies. Because the LFPs associated with 44 neurons displayed oscillations at two distinct frequency bands in the 1–10 Hz range, we were able to test if the spike–LFP phase locking was specific to an individual frequency band or present for both bands. 13.6% of these cells (6/44) showed frequency-specific phase locking, showing phase-locked firing in only one LFP frequency band (*Figure 3A*; p<0.005 in one band, p>0.1 in all other bands). In the remaining cells, 75% did not show phase-locking to any band (n = 33) or showed phase-locking to both bands (n = 5). Thus, extending previous findings (*Jacobs et al., 2007*) by examining phase-locking to adaptively-identified narrowband signals, we find that human neuronal firing can be modulated by the phase of low-frequency oscillations in a band and frequency-specific manner, as predicted by several models of neural coding (*Cohen, 2014*; *Kayser et al., 2012*; *Lisman and Jensen, 2013*; *Watrous and Ekstrom, 2014*).

## LFP-spike phase coding of goal information

To understand the behavioral relevance of phase-tuned neuronal activity, we tested whether neurons also used phase-tuned neural firing to encode spatial contextual information, analogous to the phase coding for location in the rodent hippocampus (*O'Keefe and Recce, 1993*). Our task tapped into goal-directed navigation and we therefore hypothesized that phase coding may be used to represent the patient's prospective navigational goal and should appear as neuronal firing to different phases for different navigational goals. Visual inspection of raw traces (*Figure 4—figure supplement 1*) and circular histograms of spike-phases during deliveries to each goal revealed that this pattern was evident in individual neurons (*Figure 4A–B*).

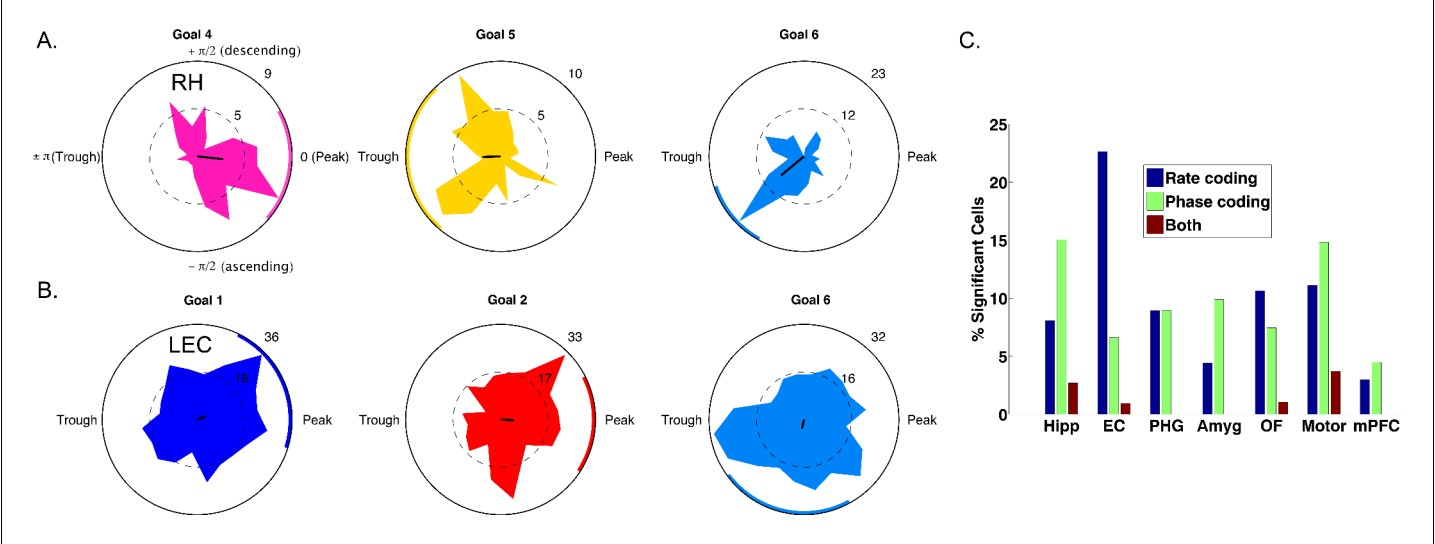

**Figure 4.** Spike–Phase coding for navigational goals. (**A**) Example neuron from the right hippocampus of patient one showing significant spike-LFP phase coding for goal four compared to goals 5 and 6. Circular histograms show spike counts separately for different goals. Black line at center of each plot shows the resultant vector and the colored arc indicates the 95th percentile confidence interval of the circular mean. (**B**) Example cell from left entorhinal cortex of patient six showing phase coding for goal 6. (**C**) Population summary showing the proportion of significant neurons in each region that showed rate coding, phase coding, or both effects. Pooling over regions, we observed significant phase coding in 10% of cells. LEC: Left entorhinal cortex; RH: Right hippocampus.

DOI: https://doi.org/10.7554/eLife.32554.008

The following figure supplements are available for figure 4:

**Figure supplement 1.** Single-trial examples of phase coding in Patient 1.

DOI: https://doi.org/10.7554/eLife.32554.009

**Figure supplement 2.** Single-trial examples of data from a neuron in Patient 11.

DOI: https://doi.org/10.7554/eLife.32554.010

We used a cross-validated decoding approach (*Watrous et al., 2015a*; see Materials and methods) to confirm that goal-specific phase variations were robust, by testing whether the patient's prospective goal could be predicted from the phase of neuronal spiking. This analysis identified 63 cells (10% of 627 cells tested) across all regions that showed individually significant decoding of goal information from spike phases (p<0.05, shuffle corrected). This proportion of neurons exceeded chance levels (Binomial test, p<0.0001) and spike phase coding differentially occurred in the hippocampus ($\chi^2(6)=50$, p<0.0001), with 28 of the phase coding cells coming from the hippocampus of nine different patients (*Figure 4C*). Roughly half (29/63) of phase coding cells exhibited significant phase-locking (Rayleigh test, p<0.005), consistent with the idea that phase-locking and phase coding are related but non-identical phenomena (*Watrous et al., 2015a*). Critically, 51 (80%) of the cells that showed significant phase coding did not show firing-rate effects (two-way ANOVA, p>0.1), indicating that phase coding for a specific goal state can exist independent of firing rate effects. We also observed intriguing examples of neurons that showed rate and phase-coding for different goals (*Figure 4—figure supplement 2*). These results indicate that rate and phase coding each contribute to the neural representation of goals during navigation.

To further understand hippocampal phase coding and motivated by our findings of differential firing rate modulation during planning and arrival, we investigated if phase coding differentially occurred during different task periods. Observing such a distinction would indicate that phase coding for goals is behaviorally relevant because the effects relate to navigational planning or goal arrival. We thus re-ran our decoding analysis for hippocampal neurons and restricted the analysis to either planning or arrival periods of each trial. We identified a subset of 24 hippocampal neurons that showed significant decoding (p<0.05) during the planning period, a proportion significantly above chance (24/186 neurons, 12% Binomial test p<0.00001, chance = 9.3 neurons). Similarly, we identified a mostly distinct subset of neurons (e.g., *Figure 4—figure supplements 1* and *2*) that

showed phase coding during goal arrival (24/186 neurons, 12%, Binomial test p<0.00001). Importantly, these were largely distinct subsets of hippocampal neurons and were found in different patient groups, with only two neurons significant for both planning and arrival. Because we imposed two statistical thresholds per cell for inclusion, this analysis is statistically more stringent than the preceding analysis, which assessed phase coding over the entire task period, and thus the neuron counts are expected to be substantially smaller.

Although our primary analysis focused on phase coding to low-frequency (1–10 Hz) oscillations because they were most prevalent in the hippocampus, we also examined whether phase coding was present at the sites that showed alpha- or beta-band oscillations (10–30 Hz). However, we did not observe phase coding beyond chance levels in this population of hippocampal cells in this band (5.2%, n = 7 of 133 neurons, chance = 5%).

In sum, we find evidence that phase-coding by individual neurons is specific to particular task periods and frequencies. We conclude that phase coding occurs in distinct hippocampal populations in support of navigational planning and goal arrival. More broadly, these findings suggest that phase-based coding to slow oscillations is a general phenomenon used by MTL neurons to represent contextual information.

We performed several control analyses to exclude alternate accounts of our findings (see Control Analyses section for full details). Average firing rates did not differ between phase-coding and non-phase-coding cells (ranksum test, p=0.39). Phase coding was not related to the overall prevalence of oscillations detected by MODAL (rho = −0.0049, p=0.9) nor to oscillatory bandwidth (rho = 0.008, p=0.85). The prevalence of oscillations was not modulated as a function of goal in the vast majority (94%) of phase-coding cells. These results indicate that phase coding for goals is more directly related to coupling between spikes and the LFP than to broader changes in LFP power or spike rates. Finally, to link these findings to our previous work using high-frequency activity (*Watrous et al., 2015a*), we observed a significant positive relationship (shuffle corrected p<0.01) between single-neuron firing rate and high-frequency activity in 41% of recorded neurons, suggesting that the phenomena are related in many cases. We thus conclude that phase coding is a robust mechanism for neural representation in the human brain during navigation.

## Discussion

Analyzing recordings from epilepsy patients performing a goal-directed navigation task, we expand our previous observation of phase-coding with high-frequency LFPs (*Watrous et al., 2015a*) to the domain of single neuron spiking. In addition to firing rate modulations (discussed below), we found a distinct population of ~10% of cells in which spike-LFP phase coding contributed to representations in the absence of significant changes in firing rate (*Hyman et al., 2005*; *Rutishauser et al., 2010*). In addition, we found neurons that were phase-locked to frequency-specific narrowband oscillations primarily in the slow-theta band. Together, these findings provide new, stronger evidence for SCERT and related models that posit a role for oscillatory phase in neural coding (*Nadasdy, 2009*; *Kayser et al., 2012*; *Lisman and Jensen, 2013*; *Watrous and Ekstrom, 2014*).

We replicated the earlier finding of firing-rate coding of goal representations in human single-cell activity (*Ekstrom et al., 2003*) and provide novel evidence for MTL and medial prefrontal neuronal firing during navigational planning (*Figure 1E*). Consistent with its role in viewpoint-dependent scene processing (*Epstein et al., 2003*), we found neurons in parahippocampal gyrus that were modulated during navigational arrival. In our analysis of goal modulation, we identified a similar number of neurons that were rate-modulated (n = 53) or spike-LFP phase modulated (n = 63) and these were largely non-overlapping neuronal populations. Phase-coding also appeared to be modulated by current task demands such that it appeared during either navigational planning or goal arrival. Because different groups of cells show rate versus phase coding for goals, it indicates that these phenomena are partially distinct (*Huxter et al., 2003*) and that phase coding is not an epiphenomenon.

Our analyses benefited from employing the MODAL algorithm, which combines features of earlier algorithms (*Whitten et al., 2011*; *Lega et al., 2012*; *Cohen, 2014*) to identify oscillatory bands in a manner that is customized for each recording site. MODAL is an improvement on these methods because it adaptively identifies oscillatory band(s) without introducing experimenter bias regarding bands of interest, excludes periods when phase is noisy because oscillations are absent, and

provides exactly one estimate of power, phase, and frequency per band and timepoint. We focused on low-frequency oscillations in this study due to the nature of our task, but it should be understood that MODAL allows one to investigate oscillatory effects such as phase-coding at higher frequency bands such as beta or gamma (*Siegel et al., 2009*; *Colgin, 2016*). Prior work has argued that the unstable shifts in gamma frequency limit their utility in phase coding (*Xing et al., 2012*). This is likely distinct from phase coding at slow frequencies in which both modeling (*Cohen, 2014*) and empirical studies (*Hutcheon and Yarom, 2000*; *Giocomo et al., 2007*) support the idea that neurons may respond maximally to inputs at particular frequencies, likely manifesting as the aggregated LFP signal (*Buzsáki et al., 2012*).

Our findings provide the first evidence of phase coding during human navigation and provide a theoretically important link to other model systems where phase coding is present (*Siegel et al., 2009*; *Kayser et al., 2009*; *Turesson et al., 2012*; *Ng et al., 2013*), such as phase-precession (*O'Keefe and Recce, 1993*; *Terada et al., 2017*). However, we also found prominent phase-locking and phase-coding to slower frequency oscillations below 5 Hz, suggesting that phase coding exists beyond the canonical 8-Hz theta signal seen in rats. These findings thus lend further credence to findings indicating that (virtual) navigation-related theta occurs at a slower frequency in humans (*Watrous et al., 2013*; *Jacobs, 2014*; *Bohbot et al., 2017*) and demonstrates that these oscillations play a functional role in modulating neuronal spiking.

Epilepsy is marked by slowing of neural oscillations which might be considered a confound in the present study. However, numerous previous studies have identified ~3-Hz oscillations in the human MTL (*Mormann et al., 2008*; *Watrous et al., 2011*; *Lega et al., 2012*; *Bush et al., 2017*), some of which had removed electrodes from the seizure onset zone or had analyzed intracranial recordings from non-epileptic patients (*Brazier, 1968*). We thus conclude that the present results would generalize to healthy populations.

These results align with work implicating the human MTL in spatial contextual representation (*Ranganath and Ritchey, 2012*) of navigational goals (*Ekstrom et al., 2003*; *Watrous et al., 2011*; *Brown et al., 2016*). Our results provide further evidence that the timing of MTL activity is critical for behavior (*Reber et al., 2017*; *Rey et al., 2014*). We speculate that the goal coding observed in this study reflects flexible coding of spatial contextual information in the service of ongoing behavior (*Warren et al., 2012*; *Yee et al., 2014*). Consistent with this interpretation, we observed cells that were phase coding either during navigational planning or goal arrival. Combined with previous human studies (*Kraskov et al., 2007*; *Lopour et al., 2013*; *Watrous et al., 2015a*; *ten Oever and Sack, 2015*), our work indicates that both firing rate and the precise timing of activity relative to LFP phase are general coding mechanisms in the human MTL across behaviors and tasks, suggesting that other types of contextual information may also be encoded using LFP phase. Future studies can build off these findings to directly assess phase coding of other types of contextual information in humans, such as phase-precession to space or time.

## Materials and methods

### Neural recordings and behavioral task

We analyzed data from 12 patients with drug-resistant epilepsy undergoing seizure monitoring (surgeries performed by I.F.). The Medical Institutional Review Board at the University of California-Los Angeles approved this study (IRB#10–000973) involving recordings from patients with drug-resistant epilepsy who provided informed consent to participate in research. Patients were implanted with microwire depth electrodes (*Fried et al., 1999*) targeting the medial temporal lobe and medial frontal lobe sites (*Figure 1—figure supplement 1*, see *Jacobs et al., 2010*; *Fried et al., 1999*; *Mukamel et al., 2010* for other example implantation images). Groups were formed for recordings in hippocampus, entorhinal cortex, parahippocampal gyrus, amygdala, orbitofrontal, (pre) motor, and cingulate/medial prefrontal cortex. (n = 282, 176, 68, 225, 200, 82, 137 neurons, respectively). Acronyms for these regions are Hipp, EC, PHG, Amy, OF, Motor, and mPFC, respectively. Subsets of these neurons were analyzed depending on the inclusion criteria for each specific analysis. For instance, only neurons with simultaneously recorded LFPs exhibiting 1–10 Hz oscillations were analyzed for phase locking and phase coding. Microwire signals were recorded at 28–32 kHz and

captured LFPs and action potentials, which were spike-sorted using *wave_clus* (*Quiroga et al., 2004*). Signals were then downsampled to 2 kHz.

We examined data from a total of 31 recording sessions in which patients performed a continuous virtual-taxi driver game in a circular environment. Patients were instructed to drive passengers to one of 6 goal stores in the virtual environment. Upon arrival, they were given a new goal destination. The task was self-paced in order to accommodate patient testing needs and therefore patients performed at ceiling. Patients performed an average of 73 deliveries in each session (standard deviation = 11 deliveries). To assess behavioral performance, we calculated the drive time for each delivery, defined as the amount of time to drive between goal stores. We binned each task session into quintiles and calculated a Kruskal-Wallis test across task sessions. The recordings and behavioral task have been detailed in prior publications that have characterized the spatial-tuning of neurons using firing rate alone (*Jacobs et al., 2010*; *Miller et al., 2015*). Here, our primary analyses focused on how contextual information about navigational goals may be encoded based on firing rates and spike-LFP interactions.

## Detection and rejection of epileptogenic signals

We implemented an automated algorithm to detect and exclude epochs of signal likely resulting from epileptic activity following prior work (*Gelinas et al., 2016*). We first low-pass filtered (fourth order Butterworth) the signal below 80 Hz to remove any spike-contamination at high frequencies. Epochs were marked for rejection if the envelope of the unfiltered signal was four standard deviations above the baseline or if the envelope of the 25–80 Hz bandpass filtered signal (after rectification) was four standard deviations above the baseline. In some cases, we noted short 'bad data' epochs lasting less than one second were not detected. We conservatively elected to exclude these epochs by marking any 'good data' epoch lasting less than one second as 'bad'. Bad data epochs were excluded from all analyses. This algorithm identified and excluded ~5% (median across LFP channels) of data. Furthermore, the rate of excluded data did not differ between the LFP channels that did and did not contain phase-coding cells (rank-sum test, p=0.9).

## Multiple oscillations detection algorithm ('MODAL')

Numerous factors contribute to the presence and characteristics of band-limited neural oscillations, broadly including neuroanatomy, behavioral state, and recording equipment (*Buzsáki et al., 2012*). We developed an algorithm to adaptively detect and characterize neural oscillations in bands exceeding the background 1/f spectrum motivated by rodent studies that exclude periods of low amplitude theta oscillations when assessing phase coding (*Lenck-Santini and Holmes, 2008*). To this end, we modified the 'frequency sliding' algorithm (*Cohen, 2014*), which provides the instantaneous phase and frequency of oscillations in a band, in two important ways.

First, rather than calculating frequency sliding in *a priori* bands, we defined bands for subsequent analysis on each electrode as those frequencies exceeding the background 1/f spectrum. We calculated power values in .5 Hz steps from 1 to 50 Hz using six cycle Morlet wavelet convolution. We then created a power spectrum by averaging values over time (and excluding bad data epochs), and fit a line to this spectrum in log-log space using *robustfit* in Matlab. Similar approaches have been used previously (*Lega et al., 2012*; *Podvalny et al., 2015*). Frequency band edges were defined as the lowest and highest frequencies in a contiguous set of frequencies that had values exceeding this fit; several bands could be detected on each electrode. We then calculated the instantaneous frequency and phase in each detected band using the 'frequency sliding' algorithm (*Cohen, 2014*).

Second, frequency sliding provides a frequency and phase estimate at every moment in time, regardless of the presence or absence of an oscillation. We ensured that phase and frequency estimates were only obtained during time periods where there was increased power in the band of interest. We recomputed the power spectrum in 10s, non-overlapping windows and recomputed the fit line as described above. We excluded phase and frequency estimates at time points (1) in which the power was below the fit line or, (2) were during bad data epochs. Finally, we also excluded noisy frequency estimates outside of the band, which can occur based on 'phase slips' (*Cohen, 2014*). MODAL was implemented in Matlab using custom code that is available on Github (https://github.com/andrew-j-watrous/MODAL; copy archived at https://github.com/elifesciences-publications/MODAL).

## Statistical analyses

To assess how neuronal activity may vary during navigational planning and goal arrival, we split each delivery in half and operationalized the first half of each delivery as the planning period and the second half of each delivery as the arrival period. This approach has the advantage of creating equally sized temporal windows for analysis but does not allow us to draw firm conclusions regarding the precise temporal dynamics of navigational planning or goal arrival. We analyzed neural firing rate using a two-way ANOVA with factors of navigational goal and task period. Cells which exhibited main effects of goal or task period (defined as p<0.05 uncorrected) were considered significant.

We used Rayleigh tests to identify phase-locked neural firing, extracting the phase of the LFP during each spike in each detected frequency band. All analyses were done considering each band separately and statistical thresholding was set at p<0.005 for each cell. This was chosen to be stricter than p<0.05 (Bonferroni-corrected) across the number of bands detected in the 1–10-Hz range. To control for the possibility that non-sinusoidal oscillations led to spurious phase-locking, we tested if the distribution of spike phases was different from the distribution of all phases on the LFP. 96% of phase-locked cells had a significantly different phase-preference to that of the entire LFP (p<0.05; Watson Williams test), suggesting that phase-locked activity was not a byproduct of non-sinusoidal oscillations.

## Assessment of phase coding using cross-validated decoding

We used a decoding-based approach to identify phase coding, employing a linear decoder with five-fold cross-validation to predict the behavioral goal from the phase of the LFP during neural spiking. In each band detected by MODAL, we first computed the sine and cosine of the phase values before classification following previous work (*Lopour et al., 2013*; *Watrous et al., 2015a*). Chance performance varies across cells because we classified goal information associated with the LFP phase for each spike and the distribution of spikes across goals varied between cells. Similarly, circular statistics can be influenced by small sample sizes. We accounted for these issues using a permutation procedure, re-running our classification 500 times per cell using shuffled goal information (*circshift* in Matlab to maintain the temporal structure of the session) to get a surrogate distribution of classification accuracies per cell. We then obtained a p-value for classification by ranking our observed classification accuracy to the surrogate distribution; p-values less than .05 were considered significant. We additionally ruled out the possibility that our phase-decoding approach was biased to observe effects in more narrow oscillatory bands, finding no correlation between phase-decoding classifier accuracy and oscillatory bandwidth (rho = -0.008, p=0.85; see also Supplemental Materials).

We then used the above decoding approach considering spikes in only the first half (planning) or second half (arrival) of each delivery to assess how phase coding varies by behavior. Each cell was categorized as phase coding during planning (p<0.05 with decoding approach), arrival (p<0.05), or both.

## Control analyses

We performed three analyses testing whether broader changes in the LFP, which may influence signal to noise ratio, may contribute to or confound our results. First, we correlated the phase-decoding classification accuracy in each band with the proportion of time oscillations were detected over the whole session. We did not observe a relation between the prevalence of oscillations and phase decoding (rho = −0.0049, p=0.9). Second, we performed an analysis testing whether phase-coding, measured by classification accuracy of our decoder, was related to the oscillatory bandwidth. We did not observe any relationship between the two measures (rho = −0.008, p=0.85), indicating that phase coding in the range we are considering (<10 Hz) is unrelated to bands with wider (possibly less stable) frequencies. Third, to determine if the prevalence of oscillations across goals could account for our phase coding results, we asked whether the prevalence of oscillatory bouts in each band varied by goal, focusing our analysis on channels where we observed phase coding cells. For each band between 1–10 Hz, we computed the prevalence of bouts for each goal. We assessed significant goal-related changes in bouts with chi-square tests, comparing the observed chi-square value to a surrogate distribution of values generated by shuffling goal information (n = 100 shuffles). We observed significant effects (p<0.05 shuffle corrected) in only 6.4% of cases which did not exceed chance levels (Binomial test, p=0.18).

To determine how the present single-neuron results relate to our previous work (*Watrous et al., 2015a*), we tested whether (1) high frequency activity (HFA; 65–120 Hz) correlated with single-neuron spiking and (2) whether HFA demonstrated phase coding to low-frequency phase. First, excluding bad epochs and analyzing the entire recording session for each LFP with an associated single-neuron recording, we correlated z-scored HFA with a smoothed firing rate vector (500 ms kernel). We identified significant (shuffle-corrected p<0.01) positive correlations between HFA and single-neuron activity in 544/1311 (41%) of neurons. In contrast, negative correlations were only identified in 14 neurons.

Second, to test whether the low-frequency phase when HFA events occurred correlated with goal state, we adapted our decoding framework to incorporate HFA events instead of spiking, using similar methods as our previous work (*Watrous et al., 2015a*). Briefly, for each channel we identified periods where the normalized HFA power exceeded z = 1.96 and labeled the center of each such period as an HFA 'event' (see Figure 2c in *Watrous et al., 2015a*). Each HFA event was then treated as if it were the timepoint of an action potential and used as the input to our main analysis of spike–phase coding. Using this approach, we found significant decoding using HFA events instead of spikes on 14% of channels (76/525), comparable to what we have previously reported (*Watrous et al., 2015a*). We then asked if the presence of HFA phase coding on a channel correlated with the existence of spike-phase coding on the same channel. However, we found that the presence of these two phenomena were unrelated across channels ($\chi^2$=0.01, p=0.9). Thus, while single-neuron firing is correlated with increases in HFA, both of which demonstrate phase coding, spike- and HFA-phase coding were not observed on the same channels. This counter-intuitive result may reflect a functional heterogeneity of information representation within the human MTL, whereby adjacent cells represent different types of information even as they contribute to the same LFP. This is similar to what has been observed in monkey orbitofrontal cortex regarding the differential information representation shown by single neurons compared to HFA (*Rich and Wallis, 2017*).

## Data availability

The raw human single-neuron recordings can be obtained upon request from Joshua Jacobs. At this point, the data has not been made publicly available to ensure controlled access to the dataset and that the patient's anonymity is not compromised.

## Acknowledgements

We wish to thank the patients for their participation in this study. This work was supported by National Institutes of Health grants NS033221 and NS084017 (IF), MH061975, MH104606 (JJ), and National Science Foundation grants GRFP (SEQ) and BCS-1724243 (JJ). We also thank the editors and three anonymous reviewers for helpful feedback that greatly improved this manuscript.

## Additional information

### Funding

| Funder | Grant reference number | Author |
| --- | --- | --- |
| National Science Foundation | DGE 16-44869 | Salman E Qasim |
| National Institute of Neurological Disorders and Stroke | NS033221 | Itzhak Fried |
| National Institute of Neurological Disorders and Stroke | NS084017 | Itzhak Fried |
| National Institute of Mental Health | MH104606 | Joshua Jacobs |
| National Science Foundation | BCS-1724243 | Joshua Jacobs |
| National Institute of Mental Health | MH061975 | Joshua Jacobs |

The funders had no role in study design, data collection and interpretation, or the decision to submit the work for publication.

## Author contributions
Andrew J Watrous, Conceptualization, Resources, Software, Formal analysis, Investigation, Methodology, Writing—original draft, Writing—review and editing; Jonathan Miller, Data curation, Software, Writing—review and editing; Salman E Qasim, Resources, Software, Writing—review and editing; Itzhak Fried, Resources, Funding acquisition, Investigation, Writing—review and editing; Joshua Jacobs, Conceptualization, Resources, Data curation, Supervision, Funding acquisition, Investigation, Visualization, Methodology, Writing—original draft, Project administration, Writing—review and editing

## Author ORCIDs
Andrew J Watrous (iD) http://orcid.org/0000-0002-3107-3726
Salman E Qasim (iD) http://orcid.org/0000-0001-8739-5962
Itzhak Fried (iD) http://orcid.org/0000-0002-5962-2678
Joshua Jacobs (iD) https://orcid.org/0000-0003-1807-6882

## Ethics
Human subjects: The Medical Institutional Review Board at the University of California-Los Angeles approved this study (IRB#10-000973) involving recordings from patients with drug-resistant epilepsy who provided informed consent to participate in research.

## Decision letter and Author response
Decision letter https://doi.org/10.7554/eLife.32554.016
Author response https://doi.org/10.7554/eLife.32554.017

# Additional files

## Supplementary files
• Transparent reporting form
DOI: https://doi.org/10.7554/eLife.32554.011

## Data availability
The raw data can be obtained upon request from Joshua Jacobs (joshua.jacobs@columbia.edu). At this point, the raw data have not been made publicly available to ensure controlled access to the dataset and that the patients' anonymity is not compromised.

The following previously published dataset was used:

| Author(s) | Year | Dataset title | Dataset URL | Database, license, and accessibility information |
|---|---|---|---|---|
| Fujisawa S, Amarasingham A, Harrison MT, Peyrache A, Buzsáki G | 2015 | Simultaneous electrophysiological recordings of ensembles of isolated neurons in rat medial prefrontal cortex and intermediate CA1 area of the hippocampus during a working memory task | http://dx.doi.org/10.6080/K01V5BWK | Publicly available at CRCNS.org - Collaborative Research in Computational Neuroscience |

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
