## [Decision Letter]

Thank you for submitting your article "Phase-tuned neuronal firing encodes human contextual representations for navigational goals" for consideration by *eLife*. Your article has been reviewed by three peer reviewers, and the evaluation has been overseen by Timothy Behrens as the Senior and Reviewing Editor. The reviewers have opted to remain anonymous.

The reviewers have discussed the reviews with one another and the Reviewing Editor has drafted this decision to help you prepare a revised submission.

Summary:

Electrophysiological studies in the rodent hippocampus have provided compelling evidence for neural phase coding during spatial navigation. Such studies observe spiking activity to be coordinated with specific phases of locomotion related theta oscillations. Observing similar coding properties in the human medial temporal lobe is exceedingly difficult. In their manuscript, Watrous et al. attempt to address this challenging question by recording spiking and LFP activity in epilepsy patients, while performing a virtual navigation task. Naturally, these data are highly novel and of great utility in establishing evidence of spike-phase coding in the human brain. The study is a clear advance on the authors paper which reported the LFP data, as they are now able to report the relationship between spiking and LFP activities. The study also proposes new methodologies that, subject to some concerns below, will likely be very useful to the field. All told, the reviewers agreed that it is an exciting and strong study.

Essential revisions:

There were many overlapping concerns amongst all three reviewers. I have tried to cluster them into themes where the reviewers are making overlapping points so that you find it easier to address the concerns together. You will see that although there was a consensus that the data are potentially extremely exciting, there was also a frustration that much of the data is hidden in the presentation, and that a number of technical issues need to be addressed.

Behaviour:

1) As a general comment, much of the results presentation is without any reference to the experimental task conditions or the anatomical sites of recording. While I appreciate the authors are trying to be concise, it does seem that talk about 'neural coding' requires clear grounding in what is being coded and where. Is it the author's key prediction that specifically the hippocampus should display spike-phase coupling during navigational goal planning, as apposed to other task conditions? Is this motivated by rodent work?

In a related comment, evidence for spike-phase coding would suggest a strong sensitivity to behavioral state, such that a critical test for the authors is comparing their analysis to different task periods, e.g. planning vs. navigation vs. arrival. These comparisons are important as the evidence shown for goal information coding is (despite looking large) a difference of ~1 spike between the preferred and non-preferred goals.

2) The link to behavior is unclear. No task description and behavioral readout is provided and the phase-coding theory could be epiphenomenal. This is particularly concerning given that most (93%) exhibit a very slow theta rhythm, which is unusual – even human studies often find theta around 6 Hz.

Technical concerns:

1) The authors introduce a new algorithm to separate oscillatory activity from 1/f fractal components. While this approach is timely and important, introducing a new algorithm requires some kind of validation that it actually works. MODAL seems to be a combination of 1/f slope fitting (e.g. like Gao and Voytek, NeuroImage) and the Cohen frequency sliding method and should be compared against established algorithms such as CGSA (coarse-grained spectral analyses, e.g. He 2010 Neuron) or IRASA (Wen and Liu 2016 BrainTopography and JNeuro). Reviewers agreed in discussion that it was important to validate the new algorithm.

2) The authors use a novel method for tracking the shifting frequency of oscillations. Is there evidence the authors can provide that supports phase-coding when the center frequency of an oscillation is unstable? In visual cortex, work by Xing et al. (2012) has argued that the variability in peak frequency and stochastic bursting nature of gamma oscillations greatly limits their coding ability. A similar critique may be applicable to the author's data.

3) The authors previously used HFA (high gamma) as a surrogate marker for spiking to demonstrate phase-coding in the MTL. Here they extend it to SUA. It would be of great interest to directly compare these metrics and to develop a better understanding how phase coding (at the population level) guides HFA, MUA and SUA. The authors could then test for how much variance is explained by theta alone (power and/or phase) as well as theta-HFA, theta-MUA and theta-SUA coupling.

4) It is unclear if there were any power differences in the theta band that might explain why some sites show more pronounced interactions. Differences in signal-to-noise could affect phase estimates.

5) Were LFPs and units extracted from the same or adjacent wires? Do the effects still hold true when the LFP is extracted from the most distal depth electrode?

6). Why was the frequency range limited to <10 Hz, which seems arbitrary given that the authors detected individual oscillations and e.g. work by the Miller group indicated a relevance of alpha/beta oscillations.

7) Was the seizure onset zone excluded or only epileptiform epochs? The Gelinas algorithm only detects sharp spikes, how did the authors deal with slowing? This could confound the 3 Hz range.

8) Was the theta oscillation sinusoidal (e.g. Cole and Voytek)? How did the authors deal with sites that had multiple low frequency peaks?

9) The difference score (DS) is not defined and it is unclear what this metric does.

Presentation of data:

1) The authors present a multitude of analyses and findings in only three figures and most analyses are not well described making it difficult to assess what was actually done. Figure 1 is merely a schematic to illustrate the oscillation detection method, Figure 2 shows summary data without providing any anatomical specificity and Figure 3 shows only a few single trial examples. Given that *eLife* does not have specific space constraints, a more careful and detailed presentation of the data would help to assess the findings.

2) Methods and results lack clarity. For example, the authors mention a decoding approach without providing any additional information on what was done or what the results were. From the description it would be impossible to replicate these analyses.

3) At several points in the Results there's a lack of specific details critical to interpretation. For example, in the last paragraph of the subsection “Phase-locked neuronal firing”, the authors mention 48 neurons that displayed two distinct oscillations. It's not clear where/why these 48 neurons are only mentioned, where they were recorded from, or what the two frequencies they are referring two. This is then followed by analysis that points to single subject data in Figure 2A, and a p-value without any reference to the test performed, and a claim of support for the SCERT model. As above, I think the author's attempt at concision has left out needed information for the reader.

4) The figures jump between data from different subjects (e.g. Figure 3) making it hard to follow an example finding across analyses for one subject, as an exemplar of the group data. Indeed much of the group data exist as p-values in text making it hard to get a sense for the effect size of the results and their across subject variability.

5) No waveforms are shown and it is unclear what kind of cells the authors isolated, how they were selected and grouped. Right now, it feels like a black-box approach and no information is provided to assess the data quality.

Further analyses:

1) How did theta differ between PFC and MTL ROIs? Was there any interaction?

2) Frontal cortex results. The statement is made that "phase coding cells were not significantly clustered by brain region". This should be clarified – so this means phase-coding cells were found in frontal cortex? If so, how many and where? Were their properties different given that the emphasis here is on MTL theta, which wouldn't be present in the non-MTL recordings.

3) It would be of great interest to assess the relationship of evoked firing vs. ongoing activity. Would one observe a theta oscillation in a spike-triggered average or is some of the very slow theta (3 Hz) driven by the stimulus (spatial navigation) or saccadic/micro-saccadic eye movements, which occur at a similar frequency. As it stands, one cannot rule out that the observed effects are solely stimulus-induced.

4) What is the relationship between the "phase coding cells" (28/158) with the phase-locked cells (n=119) ? Are the 28 a subset of the 119? More broadly, I wonder whether cells that have a phase-code could qualify as phase locked using the definition used to identify the 119 cells, since these cells (by definition) have a single preferred phase. Phase modulated cells, however, change their phase preference as a function of task. This issue would benefit from clarification.

5) Only the cells that do not exhibit a rate code for navigational code were examined. Are there also cells that have both a rate and a phase code?

Interpretation:

1) The Introduction and Discussion is too heavily focused on the "SCERT" theory – while this is certainly an interesting framework, many others have proposed similar ideas so the strong focus on this very recent "theory" is distracting and does not do the importance of this finding justice. This can be solved by more careful writing. For example, "it is unclear whether phase coding manifests in MTL neurons" is too broad of a claim, as demonstrated by the references that the authors already cite that show that MTL neurons prefer certain phases.

2) It would be helpful if the authors provide explicit predictions made by the SCERT model. I'm not sure it's clear what findings would refute their model, other than a null result of phase influence. For example, does the SCERT model make predictions about which frequencies should be influencing spiking? Or is it a more general claim that any oscillation is sufficient. Similarly, how does the model relate its framing of encoding/retrieval behavior to navigation, and specifically goal planning?

Anatomy:

1) It would be helpful if the authors made clearer reference to the anatomical sites of recording during the presentation of data. Going through the results it is often unclear where units are coming from and if they are being pool across regions.

2) No anatomical information is provided on where probes were exactly located.

3) "Frontal cortex" is too broad of a term, given the very specific frontal areas that were recorded.

[Editors' note: further revisions were requested prior to acceptance, as described below.]

Thank you for resubmitting your work entitled "Phase-tuned neuronal firing encodes human contextual representations for navigational goals" for further consideration at *eLife*. Your revised article has been favorably evaluated by Timothy Behrens (Senior and Reviewing Editor), and three reviewers.

Thanks for the substantive revisions. We all agree that they have dramatically improved the paper, but there are a number of outstanding concerns from the reviewers. I am passing these on verbatim to avoid confusing matters.

Reviewer #1:

The authors have provided a thoughtful response to reviewer comments, including several new analyses. The addition of supporting text clarifies several prior uncertainties regarding task structure, anatomical specificity and group data. I have a few remaining comments.

- The authors should qualify their reporting of phase-coding; it appears to occur in a small subset of cells, whose firing rates are sparse. Only ten percent of cells showed decodable goals from spiking phases, this should be more explicitly acknowledged throughout the manuscript. This statistic also needs to be clarified in relation to Figure 4C (caption is not clear).

- Authors note they "…were able to test if the spike-LFP phase locking was specific to an individual frequency band or present for both bands". They then report 12.5% of cells showed "frequency-specific phase locking, showing phase-locked firing in only one LFP frequency band". But it's not clear what the split is between the two frequencies for this value, and if we should infer the remaining percentage is for locking to both frequencies or none?

- The authors provide some benchmarking for the MODAL analysis technique, however, I would encourage them to pursue a separate publication of the method in the future, where the strengths and limits of the technique are rigorously quantified.

- ANOVA results are not consistently reported with degrees of freedom and F statistic – presented in some cases and not in others. This needs to be harmonized throughout the manuscript.

Reviewer #2:

The authors addressed all concerns in great detail, however, in some cases their responses fall a bit short and a few more details would help.

E.g. Technical concerns #1: applying their new algorithm to a different dataset does not constitute 'validating' it against a different established algorithm.

Technical concerns #3: While the authors demonstrate a correlation between HFA and spiking, they do not show the more obvious (which their Discussion actually implies): Phase coding as observed by Watrous et al. (2015) in *eLife* based on the HFA can also be detected based on unit activity. While the authors imply a direct link, an empirical demonstration would be more convincing. In other words, do phase-locked HFA and phase-locked spikes both support phase coding, i.e. are they separate processes or do they constitute the exact same process? This point remains unclear.

Technical concerns #5: Unfortunately, the authors state that they cannot access the macro data, which is puzzling per se, however, in many primate experiments (in particular on phase coding, e.g. Siegel et al., 2009), one typically uses phase and firing from adjacent and not the same wire.

Technical concerns #6: It seems trivial to open the band-pass up to 30 Hz and not restrict it to 10 Hz. This would allow for a better assessment of the data – in particular, since the authors show a beta band oscillation in Figure 2.

Reviewer #3:

The authors prepared an extensive and detailed revision that addresses most of the concerns I had raised. The manuscript reads much better now, and is more straightforward to understand now for me.

Beyond reproducing earlier findings, this manuscript shows two novel aspects: i) a novel theta-period detection algorithm is applied, which reveals 3-5Hz theta-frequency bouts in a number of human MTL and cortical areas. ii) it is shown that relative to these detected periods, a number of neurons phase-lock their spiking activity and the phase of this spiking activity is indicative of which goal is currently sought in a navigation task.

1) It is of some concern that the authors noted that it was not possible for them to know the seizure onset zone of the patients included. Having this information would have allowed to perform the critical control whether the same phenomena (phase locking and frequencies of detected theta periods) hold as a function of whether an electrode was inside vs. outside the seizure onset zone (i.e. to argue vs. the slowing). This information is typically easily accessible from the clinical record, so it is unclear why this was not done. Alternatively, perhaps this analysis could be done by using only neurons located on wires where the automatic "epileptic spike" algorithm did not have any hits?

2) The principle result is that for a subset of neurons, the phase of spikes is informative about the current navigation goal. It is further argued that for most such neurons, the firing rate was not informative about the current goal. However, I cannot see how it was excluded that what explains the differences in phase are aspects of the underlying LFP? i.e. the question is, for the neuron-LFP pairs for which the phase was indicative of the goal, was the LFP power used to define the phase also indicative of the goal? One argument the authors present to argue about this potential confound is to show that the number of detected oscillatory bouts does not correlate with whether a cell was phase coding (subsection “LFP-spike phase coding of goal information”). But it wasn't clear to me what exactly was measured here – please clarify. Why not specify for how many cells the detected nr bouts was indicative of the goal?

[Editors' note: further revisions were requested prior to acceptance, as described below.]

Thank you for submitting your article "Phase-tuned neuronal firing encodes human contextual representations for navigational goals" for consideration by *eLife*. Your article has been reviewed by three peer reviewers, and the evaluation has been overseen by Timothy Behrens as the Senior and Reviewing Editor. The reviewers have opted to remain anonymous.

– *Reviewer #2:*

It remains striking that the authors find that HFB and SUA are independently locked to the same theta rhythm, but are not correlated. This is surprising given their previous conclusions from Watrous (2015*eLife*) and no explanation for this finding is provided. Given that this is a direct follow-up submission, this should be discussed in more detail. In particular, given the results by Rich and Wallis (2017) and Watson (2017, European J Neurosci) it is concerning that the authors went through an elaborate analysis pipeline, which included the development of a new algorithm MODAL, to obtain these results, which are questionable if one assumes that HFB activity reflects MUA firing. What is the interpretation on the physiological level and how can this differential HFB and SUA coupling to the same theta rhythm come about? How would one reconcile these differences in terms of a mechanism?

Reviewer #3:

My remaining issues were addressed.

---

## [Author Response]

Essential revisions:There were many overlapping concerns amongst all three reviewers. I have tried to cluster them into themes where the reviewers are making overlapping points so that you find it easier to address the concerns together. You will see that although there was a consensus that the data are potentially extremely exciting, there was also a frustration that much of the data is hidden in the presentation, and that a number of technical issues need to be addressed.

We are pleased to hear enthusiasm for our manuscript. As described above, our brevity concerning data presentation was based on our earlier understanding of the format for a Research Advance in *eLife*. Now that we have been informed of the expanded length limit, we have added additional data and figures. For detail, see response to Presentation of data #1 below.

Behaviour:1) As a general comment, much of the results presentation is without any reference to the experimental task conditions or the anatomical sites of recording. While I appreciate the authors are trying to be concise, it does seem that talk about 'neural coding' requires clear grounding in what is being coded and where. Is it the author's key prediction that specifically the hippocampus should display spike-phase coupling during navigational goal planning, as apposed to other task conditions? Is this motivated by rodent work?

We apologize for the lack of detail in our original manuscript. We have now added additional analyses to further clarify the behavioral and anatomical dependencies of our data. We have added an expanded behavioral component to the manuscript by assessing neural activity during planning and goal arrival periods. We also analyze rate and phase coding separately in different MTL and frontal brain areas. These analyses show that the hippocampus is involved in navigational planning using both a rate and a phase-coding scheme.

Regarding predictions, we hypothesized that goals would be the most salient spatial contextual variable in our goal-directed navigation task and would thus be phase coded. However, we do not believe that phase coding is limited to these variables but speculate that any spatial contextual variables would also be phase-coded. We have also added the following text, which provides further detail on the key predictions in our study:

“Drawing upon the phase-coding hypotheses from SCERT and related findings in rodents (Hollup et al., 2001; Hok et al., 2007; Hyman et al., 2005; O’Neill et al., 2013), we hypothesized that spatial contextual representations for specific navigational goals would be implemented by distinctive patterns of phase coding by individual neurons. Moreover, based on rodent (Wikenheiser et al., 2015) and human studies (Viard et al., 2011; Howard et al., 2014; Brown et al., 2016; Horner et al., 2016; Bellmund et al., 2017) implicating medial temporal lobe structures and frontal cortex in navigational planning, we reasoned that spike-phase coding may support these behaviors at the single-neuron level, hypothesizing that distinctive spike phase patterns would correspond to the neural network states representing planning and searching for particular goals.”

In a related comment, evidence for spike-phase coding would suggest a strong sensitivity to behavioral state, such that a critical test for the authors is comparing their analysis to different task periods, e.g. planning vs. navigation vs. arrival. These comparisons are important as the evidence shown for goal information coding is (despite looking large) a difference of ~1 spike between the preferred and non-preferred goals.

We thank the reviewers for this helpful comment and now include an analysis which addresses the sensitivity of rate and phase coding to behavioral state. As suggested, we have added an analysis comparing phase coding during navigational planning and goal arrival. By performing this analysis, we now show that individual neurons are selectively responsive during these two task periods using rate coding (subsection “Behavior and neuronal firing during goal-directed navigational planning and arrival”) and phase coding (subsection “LFP-spike phase coding of goal information”). These findings are now described fully in the Results section and we have expanded the text regarding the behavioral relevance of our findings in the Discussion.

We believe the reviewer point regarding ~1 spike difference between the preferred and non-preferred goals is a misunderstanding. Although the difference at the maximum preferred phase is ~1 spike in the example neuron, the distributions themselves are clustered at opposite phases and are thus distinct. As we now explain more thoroughly in our revised Materials and methods section, these distinctions are captured by our cross-validated decoding approach because it would likely not detect phase coding in the case of a 1-spike difference between two distributions.

2) The link to behavior is unclear. No task description and behavioral readout is provided and the phase-coding theory could be epiphenomenal.

We now include additional analyses focused on dissecting which aspects of navigational planning or goal arrival are modulating firing rate or phase coding. As described in the new text below, we believe that this provides substantial reason to believe that the phase coding we observed is not epiphenomenal:

“In our analysis of goal modulation, we identified a similar number of neurons that were rate-modulated (n=53) or spike-LFP phase modulated (n=63) and these were largely non-overlapping neuronal populations. […] Because different groups of cells show rate versus phase coding for goals, it indicates that these phenomena are distinct (Huxter et al., 2003) and that phase coding is not an epiphenomenon.”

This is particularly concerning given that most (93%) exhibit a very slow theta rhythm, which is unusual – even human studies often find theta around 6 Hz.

We are assuming the comment regarding 6-Hz theta is in reference to human neocortical or frontal midline theta detected with EEG or MEG. While the precise nature of the human hippocampal theta rhythm remains controversial, most invasive recording studies that have investigated human memory and navigation have identified two bands centered around 3 and 8Hz (Watrous et al., 2011; Watrous et al., 2013; Vass et al., 2016; Bohbot et al., 2017; Bush et al., 2017; Agajhan et al., 2018). These studies have found modulations of neural activity particularly in the slow-theta band (~3 Hz) as a function of behavior, arguing against an epiphenomenal or epilepsy-based account of our findings. We validated the MODAL algorithm to verify it was identifying oscillations correctly by running MODAL on rodent data and by verifying its output compared to visual inspection of raw traces. To this end, we now provide more raw data (Figure 4—figure supplements 1 and 2), which demonstrate slow theta in raw traces from the human hippocampus.

Technical concerns:1) The authors introduce a new algorithm to separate oscillatory activity from 1/f fractal components. While this approach is timely and important, introducing a new algorithm requires some kind of validation that it actually works. MODAL seems to be a combination of 1/f slope fitting (e.g. like Gao and Voytek, NeuroImage) and the Cohen frequency sliding method and should be compared against established algorithms such as CGSA (coarse-grained spectral analyses, e.g. He 2010 Neuron) or IRASA (Wen and Liu 2016 BrainTopography and JNeuro). Reviewers agreed in discussion that it was important to validate the new algorithm.

We thank the reviewers for this comment, which encouraged us to validate our new algorithm. We elected to address this comment more directly by validating MODAL using rodent data, asking if MODAL can blindly detect rodent hippocampal theta, a canonical “gold standard” pattern in electrophysiology studies. We used a publicly available dataset of rodent recordings from medial prefrontal cortex and CA1 (Fujisawa et al., 2008; “pfc-2” dataset from crcns.org). As can be seen in the new figure (Figure 2—figure supplement 2), MODAL captured the 8Hz hippocampal theta rhythm and a slower and more variable rhythm in many frontal cortex channels.This analysis, coupled with visual inspection of raw traces (e.g. Figure 4—figure supplements 1 and 2) suggests that the slow theta that we observed in our human MTL recordings is not an artifact of MODAL.

2) The authors use a novel method for tracking the shifting frequency of oscillations. Is there evidence the authors can provide that supports phase-coding when the center frequency of an oscillation is unstable? In visual cortex, work by Xing et al. (2012) has argued that the variability in peak frequency and stochastic bursting nature of gamma oscillations greatly limits their coding ability. A similar critique may be applicable to the author's data.

We appreciate this feedback and performed an analysis to assess this critique of shifting frequency of oscillations empirically. We performed an analysis testing whether phase-coding, measured by classification accuracy of our decoder, was related to the oscillatory bandwidth. We did not observe any relationship between the two measures (rho = -.008, p =.85), indicating that phase coding in the range we are considering (<10 Hz) is unrelated to bands with wider (possibly less stable) frequencies and have added this to the Materials and methods and Results sections.

Theoretically, we agree that the shifting center frequency of gamma oscillations makes them a comparatively unlikely and unreliable candidate for phase coding. Instead, the SCERT model and predictions from our previous work (Watrous et al., 2015) revolve around low-frequency oscillations, which are relatively more stable compared to gamma, and we therefore chose to investigate activity below 10Hz in this study. Modeling work by Michael X Cohen and empirical work demonstrating neural resonance both support the idea that neurons may be active when there are frequency shifts in the slow oscillations. Finally, we note that the Xing et al. finding is consistent with Belitski et al. (2008) who found unique information representation in adjacent low-frequencies (below 12 Hz) but redundant information at higher frequencies. We have added the following text making these points to the Discussion:

“Prior work has argued that the unstable shifts in gamma frequency limit their utility in phase coding (Xing et al., 2012). This is likely distinct from phase coding at slow frequencies in which both modeling (Cohen, 2014) and empirical studies (Hutcheon and Yarom, 2000; Giocomo et al., 2007) support the idea that neurons may respond maximally to inputs at particular frequencies, likely manifesting as the aggregated LFP signal (Buzsaki et al., 2012).”

3) The authors previously used HFA (high gamma) as a surrogate marker for spiking to demonstrate phase-coding in the MTL. Here they extend it to SUA. It would be of great interest to directly compare these metrics and to develop a better understanding how phase coding (at the population level) guides HFA, MUA and SUA. The authors could then test for how much variance is explained by theta alone (power and/or phase) as well as theta-HFA, theta-MUA and theta-SUA coupling.

We thank the reviewer for this comment that was particularly helpful in linking the current manuscript to our prior work which used HFA. We describe our analysis in the Materials and methods and approached this issue by detecting HFA events using previous methods (Watrous et al., 2015), asking whether HFA events co-occurred with single neuron activity (SUA). We found that SUA was indeed significantly positively correlated with HFA on many cells. While we agree that the reviewer’s suggestion to comprehensively test explained variance by each neural metric could be extremely interesting, we have already performed one component of this analysis in our previous work relating theta-HFA coupling to phase-coding (Watrous et al., 2015b). We suspect that the full version of this analysis would be very challenging to conduct in our human dataset because we do not expect to reliably detect theta, SUA/MUA, and HFA on all channels (owing to the somewhat noisier recordings compared to rodent/primate signals). We believe this issue would complicate interpretation and is best addressed first in model organisms. We therefore respectfully request to defer on this issue. We have added the following text detailing our findings linking HFA to SUA to the Results:

“Finally, to link these findings to our previous work using high-frequency activity (Watrous et al., 2015), we observed a significant positive relationship (shuffle corrected p<.01) between single-neuron firing rate and high-frequency activity in 41% of neurons, suggesting that the phenomenon are related in many cases.”

4) It is unclear if there were any power differences in the theta band that might explain why some sites show more pronounced interactions. Differences in signal-to-noise could affect phase estimates.

The reviewer raises the valid point that more prevalent oscillations may be expected to be more likely to promote phase coding and conversely that sites without oscillations cannot show phase coding. Notably, we only calculate phase decoding on sites with oscillations and so our approach addresses this concern by design. We investigated this issue of variable phase estimates impacting signal-to-noise by asking if neurons that showed phase coding were recorded on LFP channels with a larger prevalence of oscillations. This addresses the reviewer concern regarding signal-to-noise because the MODAL algorithm only detects oscillations under situations of comparatively high power. We did not observe a relation between the prevalence of oscillations and phase decoding on classifier performance when we tested for this pattern by comparing mean oscillation prevalence and phase decoding rates across cells (rho = -0.0049, p=.9155). This indicates that our observations of phase-coding were not affected by the overall amount of time oscillations were detected within a band on the LFP. We have added the following text describing this analysis to the Materials and methods:

“We assessed signal to noise ratio in two ways to see if it may contribute to or confound our results. […] We did not observe any relationship between the two measures (rho = -.008, p =.85), indicating that phase coding in the range we are considering (<10 Hz) is unrelated to bands with wider (possibly less stable) frequencies.”

5) Were LFPs and units extracted from the same or adjacent wires? Do the effects still hold true when the LFP is extracted from the most distal depth electrode?

Similar to previous work, LFPs and units were extracted from the same microwire (Jacobs et al., 2007; Manning et al., 2009; Rutishauser et al., 2010; Nir et al., 2017). We unfortunately do not have access to the macro-electrode recordings and so we are unable to test how spikes relate to LFPs on the most distal macro depth electrode.

6) Why was the frequency range limited to <10 Hz, which seems arbitrary given that the authors detected individual oscillations and e.g. work by the Miller group indicated a relevance of alpha/beta oscillations.

Although we agree that there is phase coding at frequencies above 10 Hz, we chose to focus this study on slower oscillations for several reasons motivated by recent literature. First, we took inspiration from the rodent literature which has shown robust phase-coding based on theta phase precession during navigation, and reasoned that the low-frequency range would be the most likely candidate in humans. Second, we looked at this range because we primarily observed low-frequency oscillations and prior studies have observed phase-locking to this band (Jacobs et al., 2007). Third, the SCERT model posits that low-frequency oscillations will modulate neural firing and several studies have reported theta oscillations in the human MTL during navigation (Watrous et al., 2011; Watrous et al., 2013; Bohbot et al., 2017; Vass et al., 2017; Bush at et al., 2017; Aghajan et al., 2018). Finally, we wish to note that the apha or beta frequencies reported by the Miller group were generally found in the neocortex, rather than the MTL where the bulk of our electrode coverage was present. We have added the following text to the Discussion:

“We focused on low-frequency oscillations in this study due to the nature of our task, but it should be understood that MODAL allows one to investigate oscillatory effects such as phase-coding at higher frequency bands such as beta or gamma (Siegel et al., 2009; Colgin et al., 2016).”

7) Was the seizure onset zone excluded or only epileptiform epochs? The Gelinas algorithm only detects sharp spikes, how did the authors deal with slowing? This could confound the 3 Hz range.

We did not exclude the seizure onset zone in this study because we unfortunately do not have access to this information. The reviewer is correct that the Gelinas algorithm only detects sharp spikes and not possible slowing. While we cannot definitively rule out slowing, we believe several factors argue against epilepsy accounting for our results, First, epilepsy is unlikely to impact neural activity in a task-specific manner, as in the spike-phase coding to particular navigational goals that we report here. Second, our prior work (Watrous et al., 2011; Watrous et al., 2015) and that of several other groups have observed similar behavioral and neural results (including ~3 Hz oscillations) when including or excluding the seizure onset zone. Third, we now provide example raw traces (e.g. Figure 4—figure supplements 1 and 2) demonstrating phase coding to rhythmic ~3Hz oscillations. Finally, MEG studies have also source-localized ~3 Hz activity to the MTL (Staudigl et al., 2013; Backus et al., 2016; Dalal et al., 2013). We thus conclude that our findings cannot be attributed solely to epilepsy. We have added a concise version of this argument to the Discussion addressing these issues:

“Epilepsy is marked by slowing of neural oscillations which might be considered a confound in the present study. […] We thus conclude that the present results would generalize to healthy populations.”

8) Was the theta oscillation sinusoidal (e.g. Cole and Voytek)?

Whereas some reports have noted non-sinusoidal oscillations in humans (Cole and Voytek 2017; Vaz et al., 2017), we remain agnostic to this issue in the manuscript because it is unknown if non-sinusoidal or sinusoidal oscillations impact phase coding, an issue that is still being delineated in the literature. Instead, we believe that the fundamental issue is whether non-sinusoidal oscillations contribute to spurious phase-locking and developed an approach to assess this possibility. If the oscillation is not sinusoidal, then the phase distribution over time is not uniform and may lead to spurious phase-locking. To address this, we compared the phase distribution of the entire recording to the distribution of spike phases using Watson-Williams tests (equivalent to circular t-test) for circular means. We found that nearly all of our phase-locked cells were locked to a different phase than the LFP phase across all time points, indicating that if the theta oscillation was not sinusoidal, it did not bias our phase-locking results. We have added text describing this to the Materials and methods section as follows:

“To control for the possibility that non-sinusoidal oscillations led to spurious phase-locking, we tested if the distribution of spike phases was different from the distribution of all phases on the LFP. 96% of phase-locked cells had a significantly different phase-preference to that of the entire LFP (p<.05; Watson Williams test), suggesting that phase-locked activity was not a byproduct of non-sinusoidal oscillations.”

How did the authors deal with sites that had multiple low frequency peaks?

MODAL identifies all narrowband peaks exceeding the 1/f spectrum and can explicitly identify situations in which there are multiple low-frequency peaks. For this reason, all statistics are computed on a band-by-band basis with appropriate correction for multiple comparisons. In cases in which multiple low-frequency peaks were detected (e.g. Figure 2A) we were able statistically test for phase locking and/or coding in each band. We have clarified this issue in the text as follows:

“The SCERT model predicts that neuronal activity is modulated by oscillations at particular frequencies. Because the LFPs associated with 48 neurons displayed oscillations at two distinct frequency bands in the 1–10-Hz range, we were able to test if the spike–LFP phase locking was specific to an individual frequency band or present for both bands.”

9) The difference score (DS) is not defined and it is unclear what this metric does.

We have removed all mention of difference scores from the manuscript. Instead, we now use a cross-validated decoding based approach throughout the manuscript when discussing phase coding, which is more straightforward to understand and explains the same core scientific issue.

Presentation of data:1) The authors present a multitude of analyses and findings in only three figures and most analyses are not well described making it difficult to assess what was actually done. Figure 1 is merely a schematic to illustrate the oscillation detection method, Figure 2 shows summary data without providing any anatomical specificity and Figure 3 shows only a few single trial examples. Given that eLife does not have specific space constraints, a more careful and detailed presentation of the data would help to assess the findings.

Our original understanding of this article type was that there is a strict length constraint, but this was corrected by the editorial office who confirmed there is no constraint for our revision. As such, we have added a substantial series of new figures and data analyses to further detail our findings and provide the reader with a better sense of the data. These new figures are as follows:

- Added Figure 1 showing firing rate effects during navigational planning and goal arrival;

- Added Figure 3D showing phase-locking in different brain areas;

- Added Figure 4 showing phase coding examples and a population summary for rate coding, phase coding, and both types of coding in each brain area;

- Added Figure 1—figure supplement 1 showing a task schematic, an example patient implantation image, and spike waveforms;

- Added Figure 2—figure supplement 1 showing the proportion of channels in different brain regions with detected oscillations using MODAL;

- Added Figure 2—figure supplement 2 showing the results of using MODAL on rodent data;

- Added Figure 4—figure supplements 1 and 2 showing example raw traces and phase histograms for a phase coding neuron during goal arrival (1) and a rate and phase coding neuron (2).

2) Methods and results lack clarity. For example, the authors mention a decoding approach without providing any additional information on what was done or what the results were. From the description it would be impossible to replicate these analyses.

We have expanded the Materials and methods section substantially in this revision, including new details on anatomy, statistical analyses, and decoding. In particular, we also include an expanded description of our decoding approach, copied below:

“We used a decoding-based approach to identify phase coding, employing a linear decoder with fivefold cross-validation to predict the behavioral goal from the phase of the LFP during neural spiking. […] We accounted for these issues using a permutation procedure, re-running our classification 500 times per cell using shuffled goal information (*circshift* in Matlab to maintain the temporal structure of the session) to get a surrogate distribution of classification accuracies per cell.”

3) At several points in the Results there's a lack of specific details critical to interpretation. For example, in the last paragraph of the subsection “Phase-locked neuronal firing”, the authors mention 48 neurons that displayed two distinct oscillations. It's not clear where/why these 48 neurons are only mentioned, where they were recorded from, or what the two frequencies they are referring two. This is then followed by analysis that points to single subject data in Figure 2A, and a p-value without any reference to the test performed, and a claim of support for the SCERT model. As above, I think the author's attempt at concision has left out needed information for the reader.4) The figures jump between data from different subjects (e.g. Figure 3) making it hard to follow an example finding across analyses for one subject, as an exemplar of the group data. Indeed much of the group data exist as p-values in text making it hard to get a sense for the effect size of the results and their across subject variability.

We have greatly expanded the relevant sections of text in the manuscript and have added several figures and now hope the reviewer will be satisfied with the level of detail which we provide. We identified rate coding in 11 of 12 patients and phase coding in the hippocampus of 9 patients and have added these details to the Results section.

5) No waveforms are shown and it is unclear what kind of cells the authors isolated, how they were selected and grouped. Right now, it feels like a black-box approach and no information is provided to assess the data quality.

We now include example spike waveforms in Figure 1—figure supplement 1, provide information about spike sorting and anatomical groping in the Materials and methods section, and show example raw data in Figure 4—figure supplements 1 and 2.

Further analyses:1) How did theta differ between PFC and MTL ROIs? Was there any interaction?

We now include the results of applying the MODAL algorithm for each of the 7 regions analyzed in this study in Figure 2—figure supplement 1. This analysis shows that ~3-Hz theta was observed on a similar proportion of channels in each brain area, thus confirming our focus on this range across our data.

2) Frontal cortex results. The statement is made that "phase coding cells were not significantly clustered by brain region". This should be clarified – so this means phase-coding cells were found in frontal cortex? If so, how many and where? Were their properties different given that the emphasis here is on MTL theta, which wouldn't be present in the non-MTL recordings.

We have clarified this issue throughout the manuscript, and based on other feedback (see also Further analyses #5 below), we have modified our approach for identifying phase coding neurons. In our new analysis, we now characterize phase coding as a function of brain region and find that phase coding is most prominent in the hippocampus but also occurs in frontal areas (Figure 4C). We focused on MTL results because they were the most pronounced and because we had the strongest predictions in this brain area. As indicated by the reviewer, the phase coding we observe in frontal cortex is indeed likely related to frontal theta oscillations, as we did not investigate inter-areal phase synchronization or spike-field coherence (e.g. Voytek et al., 2015; Hyman et al., 2005; Benchenane et al., 2010) in this study. We believe this is an exciting avenue for future human studies.

3) It would be of great interest to assess the relationship of evoked firing vs. ongoing activity. Would one observe a theta oscillation in a spike-triggered average or is some of the very slow theta (3 Hz) driven by the stimulus (spatial navigation) or saccadic/micro-saccadic eye movements, which occur at a similar frequency. As it stands, one cannot rule out that the observed effects are solely stimulus-induced.

Given the continuous nature of our navigation task, we are a bit unclear as to what the reviewer is referring to here regarding evoked activity. Nonetheless, we have attempted to link our findings to behavior more carefully through analyses of neural activity related to navigational planning and goal arrival. As seen in Figure 2A, we do indeed see rhythmicity in the spike-triggered average of phase-locked cells. We have also observed such 3Hz oscillations in previous navigation datasets (Watrous et al., 2013; Hippocampus). As we did not record eye-movements, we are unable to tease apart changes in theta related to navigation versus eye movements.

4) What is the relationship between the "phase coding cells" (28/158) with the phase-locked cells (n=119) ? Are the 28 a subset of the 119? More broadly, I wonder whether cells that have a phase-code could qualify as phase locked using the definition used to identify the 119 cells, since these cells (by definition) have a single preferred phase. Phase modulated cells, however, change their phase preference as a function of task. This issue would benefit from clarification.

We thank the reviewer for the feedback and recognize that the distinction between “phase-locked” and “phase-coding” is nuanced and we have refined the text to more clearly differentiate these phenomena. We now more clearly explain how phase-locking and phase-coding are not inherently the same phenomena and we also describe these two phenomena separately. We report that roughly half (29/63) of our phase coding cells showed significant phase-locking (Rayleigh test, p<.005). We have added the following text describing and interpreting these findings to the Results section:

“Roughly half (29/63) of phase coding cells exhibited significant phase-locking (Rayleigh test, p< 0.005), consistent with the idea that phase-locking and phase coding are related but non-identical phenomena.”

5) Only the cells that do not exhibit a rate code for navigational code were examined. Are there also cells that have both a rate and a phase code?

Based on this feedback, we have overhauled our analysis and now analyze rate coding and phase coding in all cells and then look for cells which show both types of effects (Figure 4C). There are indeed cells that show both rate and phase coding and we have included an example cell showing rate coding for Goal 3 and phase coding for Goal 1 showing this phenomenon in Figure 4—figure supplement 2. However, as can be seen in Figure 4C, most cells do not show both types of coding. We note that neurons which show both rate and phase coding are expected to be more rare based on the more stringent inclusion criteria (p<.05 for both rate and phase coding) and we therefore do not draw strong conclusions about the paucity of neurons showing rate and phase-coding.

Interpretation:1) The Introduction and Discussion is too heavily focused on the "SCERT" theory – while this is certainly an interesting framework, many others have proposed similar ideas so the strong focus on this very recent "theory" is distracting and does not do the importance of this finding justice. This can be solved by more careful writing. For example, "it is unclear whether phase coding manifests in MTL neurons" is too broad of a claim, as demonstrated by the references that the authors already cite that show that MTL neurons prefer certain phases.

As suggested, we have broadened the Introduction and conclusion throughout the manuscript. We did not at all intend to imply that SCERT is the first or only model to posit phase coding, and owing to the expanded length limit we are now able to explain our view on this point in a more thorough fashion. We nonetheless still make a number of references to SCERT in the paper, which we feel is justified because the Research Advance format is designed to provide advances compared to a previous *eLife* paper.

Regarding the specific example raised, we now explain the extant literature in relation to our hypotheses in more detail. The revised text reads as follows:

“However, given the complex and variable relationship (Ekstrom et al., 2007; Manning et al., 2009; Rey et al., 2014) between the spiking of particular single neurons and high-frequency activity in the human medial temporal lobe (MTL), it is unclear whether human MTL neurons show phase coding of navigationally relevant information beyond an overall preference to fire at particular phases (Jacobs et al., 2007).”

2) It would be helpful if the authors provide explicit predictions made by the SCERT model. I'm not sure it's clear what findings would refute their model, other than a null result of phase influence. For example, does the SCERT model make predictions about which frequencies should be influencing spiking? Or is it a more general claim that any oscillation is sufficient. Similarly, how does the model relate its framing of encoding/retrieval behavior to navigation, and specifically goal planning?

Regarding the issue of framing encoding/retrieval to navigation, it is in principle straightforward to relate encoding to environmental learning through navigation (e.g. “driving”) in the first trials of a task and retrieval to “planning”. Studies which have taken this approach typically have many more trials than we have available in this dataset and so we are unable to perform such an analysis here. Nonetheless, SCERT does make specific predictions regarding the frequencies which should show phase coding and we have added the following text to the Introduction which outlines our specific hypotheses (see Behaviour #1 above for further text regarding specific predictions):

“SCERT generally predicts that oscillatory frequencies should match between encoding and retrieval and that phase coding should occur at the dominant oscillatory frequency that occurs in a particular behavior and brain region. Thus, based on the body of evidence indicating hippocampal slow-theta oscillations are the most prominent during human virtual navigation (Ekstrom et al., 2005; Watrous et al., 2011; Jacobs 2014), we predicted here that phase coding should occur primarily at slow theta frequencies.”

Anatomy:1) It would be helpful if the authors made clearer reference to the anatomical sites of recording during the presentation of data. Going through the results it is often unclear where units are coming from and if they are being pool across regions.2) No anatomical information is provided on where probes were exactly located.3) "Frontal cortex" is too broad of a term, given the very specific frontal areas that were recorded.

We apologize for the vague terminology and lack of reference to anatomical sites. Frontal recordings were primarily located in orbitofrontal, premotor, and medial prefrontal/cingulate cortex. We have refined our terminology to reflect this throughout the manuscript. We also explicitly reference the locations of exemplar responses in the main text of the manuscript instead of putting this information in the figure or caption. Finally, we have added a co-registered CT-MRI implantation image to Figure 1—figure supplement 1 which can help the reader understand the recording scheme.

[Editors' note: further revisions were requested prior to acceptance, as described below.]

Reviewer #1:The authors have provided a thoughtful response to reviewer comments, including several new analyses. The addition of supporting text clarifies several prior uncertainties regarding task structure, anatomical specificity and group data. I have a few remaining comments.- The authors should qualify their reporting of phase-coding; it appears to occur in a small subset of cells, whose firing rates are sparse. Only ten percent of cells showed decodable goals from spiking phases, this should be more explicitly acknowledged throughout the manuscript. This statistic also needs to be clarified in relation to Figure 4C (caption is not clear).

We have qualified our reporting of the cells that showed phase coding in the discussion, as suggested. As suggested, we also now explicitly test the idea that the cells that exhibit phase coding have significantly lower firing rates, but did not find evidence of such a pattern. We now include this in the Results as follows:

“Average firing rates did not differ between phase-coding and non-phase-coding cells (ranksum test, p= 0.39).”

As requested, we have also clarified this information in the caption of Figure 4C.

- Authors note they "…were able to test if the spike-LFP phase locking was specific to an individual frequency band or present for both bands". They then report 12.5% of cells showed "frequency-specific phase locking, showing phase-locked firing in only one LFP frequency band". But it's not clear what the split is between the two frequencies for this value, and if we should infer the remaining percentage is for locking to both frequencies or none?

We have updated the wording of this section to now explicitly state the percentages of neurons that showed phase locking to zero, one, or more bands. The revised text reads as follows:

“Because the LFPs associated with 44 neurons displayed oscillations at two distinct frequency bands in the 1–10-Hz range, we were able to test if the spike–LFP phase locking was specific to an individual frequency band or present for both bands. 13.6% of these cells (6/44) showed frequency-specific phase locking, showing phase-locked firing in only one LFP frequency band (Figure 3A; p<.005 in one band, p>.1 in all other bands). In the remaining cells, 75% did not show phase-locking to any band (n=33) or showed phase-locking to both bands (n=5).”

Please note that we have also fixed a small typo in this section, where we previously reported 48 cells it now correctly reads 44. Our conclusions remain the same.

- The authors provide some benchmarking for the MODAL analysis technique, however, I would encourage them to pursue a separate publication of the method in the future, where the strengths and limits of the technique are rigorously quantified.

We agree with this suggestion and intend to write a separate methods manuscript on MODAL in the future.

- ANOVA results are not consistently reported with degrees of freedom and F statistic – presented in some cases and not in others. This needs to be harmonized throughout the manuscript.

We have reviewed the text carefully and made edits to ensure our statistical reporting is fully consistent.

Reviewer #2:The authors addressed all concerns in great detail, however, in some cases their responses fall a bit short and a few more details would help.E.g. Technical concerns #1: applying their new algorithm to a different dataset does not constitute 'validating' it against a different established algorithm.

We have changed the phrasing here and now refer to this testing as “benchmarking” the algorithm, which was the terminology suggested by reviewer 1.

Technical concerns #3: While the authors demonstrate a correlation between HFA and spiking, they do not show the more obvious (which their Discussion actually implies): Phase coding as observed by Watrous et al. (2015) in eLife based on the HFA can also be detected based on unit activity. While the authors imply a direct link, an empirical demonstration would be more convincing. In other words, do phase-locked HFA and phase-locked spikes both support phase coding, i.e. are they separate processes or do they constitute the exact same process? This point remains unclear.

We answer this question by reporting new analyses that directly compare phase coding via spiking and/or HFA on the same channels. This analysis showed that there was no correlation between the presence of spike and HFA phase coding on individual channels, thus indicating that the neuronal correlates of these two phenomena are distinct and, moreover, that one is not an artifact of the other. The revised text, which is in the Control Analyses section, reads as follows:

“To determine how the present single-neuron results relate to our previous work (Watrous et al., 2015b), we tested whether (1) high frequency activity (HFA; 65–120 Hz) correlated with single-neuron spiking and (2) whether HFA demonstrated phase coding to low-frequency phase. […] We then asked if the presence of HFA phase coding on a channel correlated with the existence of spike-phase coding on the same channel. However, we found that the presence of these two phenomena were unrelated across channels (χ^2^=.01, p=.9).”

This new analysis addresses the reviewer’s concern by showing that spike and HFA phase coding do not constitute the exact same process. Nonetheless, more work is necessary to understand the spatial scale of different phase coding schemes in the brain and explain why some such patterns are visible in HFA signals.

Technical concerns #5: Unfortunately, the authors state that they cannot access the macro data, which is puzzling per se, however, in many primate experiments (in particular on phase coding, e.g. Siegel et al., 2009), one typically uses phase and firing from adjacent and not the same wire.

We think that any potential concern about artifactual results from measuring spikes and LFPs on the same channels is mitigated by our new analysis above (Technical concerns #3), which shows that our observation of spike phase coding on an individual neuron is uncorrelated with the presence of HFA phase coding on the same channel. This suggests it is unlikely that our results are a result of cross-frequency “spillover” as discussed in the Siegel et al. (2009) paper.

Technical concerns #6: It seems trivial to open the band-pass up to 30 Hz and not restrict it to 10 Hz. This would allow for a better assessment of the data – in particular, since the authors show a beta band oscillation in Figure 2.

We agree and thank the reviewer for this suggestion, which has clarified the frequency specificity of phase coding in the hippocampus. We now assess hippocampal phase coding for oscillations up to 30Hz, but we did not find significant counts of neurons showing phase coding of goal information for oscillations in the alpha-beta range (10-30Hz, n = 7 of 133 neurons, 5.2%). We have included this analysis in the Results section as follows:

“Although our primary analysis focused on phase coding to low-frequency (1-10 Hz) oscillations because they were most prevalent in the hippocampus, we also examined whether phase coding was present at the sites that showed alpha- or beta-band oscillations (10–30 Hz). However, we did not observe phase coding beyond chance levels in this population of hippocampal cells in this band (5.2%, n=7 of 133 neurons, chance = 5%).”

Reviewer #3:The authors prepared an extensive and detailed revision that addresses most of the concerns I had raised. The manuscript reads much better now, and is more straightforward to understand now for me.Beyond reproducing earlier findings, this manuscript shows two novel aspects: i) a novel theta-period detection algorithm is applied, which reveals 3-5Hz theta-frequency bouts in a number of human MTL and cortical areas. ii) it is shown that relative to these detected periods, a number of neurons phase-lock their spiking activity and the phase of this spiking activity is indicative of which goal is currently sought in a navigation task.1) It is of some concern that the authors noted that it was not possible for them to know the seizure onset zone of the patients included. Having this information would have allowed to perform the critical control whether the same phenomena (phase locking and frequencies of detected theta periods) hold as a function of whether an electrode was inside vs. outside the seizure onset zone (i.e. to argue vs. the slowing). This information is typically easily accessible from the clinical record, so it is unclear why this was not done. Alternatively, perhaps this analysis could be done by using only neurons located on wires where the automatic "epileptic spike" algorithm did not have any hits?

We agree that it is unfortunate that we do not have access to the clinical records for these patients. However, we performed a new analysis to examine this issue, closely following the reviewer’s suggestion.

We used the epileptic-spike detection algorithm (Gelinas et al., 2016) to exclude epileptic events on each channel. Next, we tested whether the proportions of such events differed for channels that did or did not contain neurons showing phase coding. This analysis revealed no difference in the prevalence of epileptic events between the two groups of channels (rank-sum test, p=.9). The revised text reads as follows:

“This algorithm identified and excluded ~5% (median across LFP channels) of data. Furthermore, the rate of excluded data did not differ between the LFP channels that did and did not contain phase-coding cells (rank-sum test, p=.9).”

2) The principle result is that for a subset of neurons, the phase of spikes is informative about the current navigation goal. It is further argued that for most such neurons, the firing rate was not informative about the current goal. However, I cannot see how it was excluded that what explains the differences in phase are aspects of the underlying LFP? i.e. the question is, for the neuron-LFP pairs for which the phase was indicative of the goal, was the LFP power used to define the phase also indicative of the goal? One argument the authors present to argue about this potential confound is to show that the number of detected oscillatory bouts does not correlate with whether a cell was phase coding (subsection “LFP-spike phase coding of goal information”). But it wasn't clear to me what exactly was measured here – please clarify.

The revised manuscript addresses the potential confounds associated with changes in power in two ways. First, the existing analysis, now clarified in the first paragraph of the subsection “Control Analyses”, tested whether the number of oscillatory bouts on a channel correlates with our ability to detect phase coding at that site. By showing no effect, we rule out a signal-to-noise confound. Second, in a new analysis described below, we ask whether goal states modulate the prevalence of oscillations, which may possibly give rise to our observation of phase coding. We also rule out this possibility. Together, these two analyses indicate that spike-LFP phase coding, rather than power changes in the LFP, is the strongest indicator of goal states in our recordings.

Why not specify for how many cells the detected nr bouts was indicative of the goal?

As suggested, we tested if the prevalence of oscillations varied by goal for the neurons that exhibited phase coding. However, we found that the number of neuron–band pairs that showed power changes for individual goals did not exceed chance levels (6.4%, 6/93 significant pairs, chance = 5%). We have included a reference to these new results in the Results section and fully describe the new analysis in the Control Analyses section. Notably, this observation is broadly consistent with our previous findings on power modulations related to goal states (Watrous et al., 2011).

Results text:

“The prevalence of oscillations was not modulated as a function of goal in the vast majority (94%) of phase-coding cells. These results indicate that phase coding is more directly related to coupling between spikes and the LFP, rather than broader changes in LFP power or spike rates.”

Control Analysis text:

“Third, to determine if the prevalence of oscillations across goals could account for our phase coding results, we asked whether the prevalence of oscillatory bouts in each band varied by goal, focusing our analysis on channels where we observed phase coding cells. […] We observed significant effects (p<.05 shuffle corrected) in only 6.4% of cases which did not exceed chance levels (Binomial test, p =.18).

[Editors' note: further revisions were requested prior to acceptance, as described below.]

Reviewer #2:It remains striking that the authors find that HFB and SUA are independently locked to the same theta rhythm, but are not correlated. This is surprising given their previous conclusions from Watrous (2015) and no explanation for this finding is provided. Given that this is a direct follow-up submission, this should be discussed in more detail. In particular, given the results by Rich and Wallis (2017) and Watson (2017, European J Neurosci) it is concerning that the authors went through an elaborate analysis pipeline, which included the development of a new algorithm MODAL, to obtain these results, which are questionable if one assumes that HFB activity reflects MUA firing. What is the interpretation on the physiological level and how can this differential HFB and SUA coupling to the same theta rhythm come about? How would one reconcile these differences in terms of a mechanism?

We thank the reviewer for bringing these papers to our attention and apologize for the lack of clarity regarding this point. To summarize our relevant findings:

1) Spiking and HFA increases are correlated on 41% of channels;

2) Spikes show phase coding;

3) HFA shows phase coding;

4) Channels with spike-phase coding are not correlated with those showing HFA-phase coding.

Our findings that the same electrodes do not show both spike-phase and HFA-phase coding is consistent with the above study by Rich and Wallis, in which they report that, “Although there was a good deal of similarity between neurons and HGA encoding at the population level, this did not extend to hyperlocal pairings of neurons and HGA recorded from the same electrode. The variable(s) encoded by a given neuron did not predict those encoded by the corresponding HGA.”

We have added the following text to the Control Analysis section that clarifies these points:

“Thus, while single-neuron firing is correlated with increases in HFA, both of which demonstrate phase coding, spike- and HFA-phase coding were not observed on the same channels. […] This is similar to what has been observed in monkey orbitofrontal cortex regarding the differential information representation shown by single neurons compared to HFA (Rich and Wallis, 2017).”